



# Powering aircraft with 100% sustainable aviation fuel reduces ice crystals in contrails

Raphael Satoru Märkl[1, 2], Christiane Voigt[1, 2], Daniel Sauer[1], Rebecca Katharina Dischl[1, 2], Stefan Kaufmann[1], Theresa Harlaß[1], Valerian Hahn[1, 2], Anke Roiger[1], Cornelius Weiß-Rehm[1], Ulrike Burkhardt[1], Ulrich Schumann[1], Andreas Marsing[1], Monika Scheibe[1], Andreas Dörnbrack[1], Charles Renard[3], Maxime Gauthier[3], Peter Swann[4], Paul Madden[4], Darren Luff[4], Reetu Sallinen[5], Tobias Schripp[6], and Patrick Le Clercq[6]

[1]Deutsches Zentrum für Luft- und Raumfahrt, Institut für Physik der Atmosphäre, Oberpfaffenhofen, Germany
[2]Johannes Gutenberg-Universität, Institute of Atmospheric Physics, Mainz, Germany
[3]Airbus Operations SAS, Toulouse, France
[4]Rolls-Royce plc., Derby, UK
[5]Neste Corporation, Innovation, Porvoo, Finland
[6]Deutsches Zentrum für Luft- und Raumfahrt, Institute of Combustion Technology, Stuttgart, Germany

**Correspondence:** Raphael Märkl (raphael.maerkl@dlr.de)

**Abstract.** Powering aircraft by sustainable aviation fuels (SAF) is a pathway to reduce the climate impact of aviation by lowering aviation life-cycle $CO_2$ emissions and by reducing ice crystal numbers and radiative forcing from contrails. While the effect of SAF blends on contrails has been measured previously, here we present novel measurements on particle emission and contrails from 100% SAF combustion. During the ECLIF3 (Emission and CLimate Impact of alternative Fuels) campaign, a collaboration between DLR, AIRBUS, ROLLS-ROYCE and NESTE, the DLR Falcon 20 research aircraft performed in situ measurements following an Airbus A350-941 source aircraft powered by Rolls-Royce Trent XWB-84 engines in 1 to 2 min old contrails at cruise altitudes. Apparent ice emission indices of 100% HEFA-SPK (Hydro-processed Esters and Fatty Acids - Synthetic Paraffinic Kerosene) were measured and compared to Jet A-1 fuel contrails at similar engine and ambient ice-supersaturated conditions within a single flight. A 56% reduction of ice particle numbers per mass of burned fuel was measured for 100% HEFA-SPK compared to Jet A-1 at engine cruise conditions. The measured 35% reduction in soot particle numbers suggest reduced ice activation by the low sulfur HEFA fuel. Contrail properties are consistently modelled with a contrail plume model. Global climate model simulations for the 2018 fleet conservatively estimate a 26% decrease in contrail radiative forcing and stronger decreases for larger particle reductions. Our results indicate that higher hydrogen content fuels as well as clean engines with low particle emissions may lead to reduced climate forcing from contrails.

## 1 Introduction

As a rapidly growing industry on a global scale, aviation is faced with the challenge of being able to grow in a way that is sustainable and in line with the goals of the Paris Climate Agreement (Grewe et al., 2021). Besides the contribution of carbon dioxide ($CO_2$) to aviation climate forcing, non-$CO_2$ effects from contrail cirrus and nitrogen oxides play a major role



in aviation's effective radiative forcing (ERF), amounting to a globally averaged climate forcing contribution share of ~3.5%
for aviation (Lee et al., 2021). Contrail cirrus can locally either be cooling or warming depending on the angle of incoming
solar radiation (Meerkötter et al., 1999; Stuber et al., 2006; Forster et al., 2012; Teoh et al., 2022a). Additionally, the contrail
cirrus cover has a diurnal cycle, which is dependent on the frequency distribution of contrail cirrus life times (Newinger and
Burkhardt, 2012), as well as seasonal and regional variability (Meijer et al., 2022). The global net radiative effect however is
predicted to be warming, amounting to an ERF of 57 $\mathrm{mW/m^2}$ for the year 2018 (Lee et al., 2021). The contrail cirrus ERF
constitutes the largest contribution to aviation's total net ERF while accumulated aviation carbon dioxide emissions account
for 34 $\mathrm{mW/m^2}$ and nitrogen oxide emission for 17 $\mathrm{mW/m^2}$ (Lee et al., 2021). Furthermore, global flight distance is projected
to increase by a factor of four until 2050 compared to 2006 (Chen and Gettelman, 2016), leading to an expected tripling of
contrail cirrus radiative forcing (Bock and Burkhardt, 2019) when assuming the four-fold increase in air traffic projected by
the AEDT air traffic inventory (Wilkerson et al., 2010). In contrast, several studies have shown that the radiative forcing from
contrails was significantly reduced in the COVID-19 related low air traffic scenario (Quaas et al., 2021; Schumann et al., 2021;
Gettelman et al., 2021; Duda et al., 2023; Teoh et al., 2023a).

Combustion products from aircraft engines consist mainly of carbon dioxide ($CO_2$) and water vapor ($H_2O$), which are emitted
together with nitrogen oxides, sulfate precursors, a number of unburned organic compounds, soot and volatile aerosol particle
precursors. These hot engine emissions mix with ambient air turbulently and rapidly cool down so that the plume surpasses
water saturation and the water vapor condenses preferably onto emitted soot particles (Kärcher and Yu, 2009; Wong and
Miake-Lye, 2010; Bier et al., 2022). These small water droplets continue to grow from the surrounding available water vapor
and freeze instantaneously at cold temperatures. Hence, contrail ice crystals contain ice nuclei and therefore supercooled water
droplets are not present after the first seconds of formation. Whether contrails form or not is governed by ambient temperature,
pressure, and humidity as well as engine and fuel parameters as described by the Schmidt-Appleman threshold temperature
($T_{SA}$) (Schumann, 1996). Following ice crystal formation, the young contrail ice particles can either sublimate quickly or grow
into contrail cirrus clouds that can remain in the atmosphere for several hours in ice-supersaturated air (Minnis et al., 1998;
Jensen et al., 1998; Lewellen, 2014; Vázquez-Navarro et al., 2015; Unterstrasser et al., 2017). While $CO_2$ can remain in the
atmosphere for centuries, contrail cirrus have lifespans of several hours, which enables an immediate ERF reduction potential
when avoiding or decreasing contrail formation. Optimizing flights to avoid warming contrails by horizontal or vertical flight
route adaptation based on contrail prediction models is one possible climate impact mitigation strategy under investigation
(Schumann et al., 2011; Grewe et al., 2017; Teoh et al., 2020a, b, 2023a). However, increased fuel use and $CO_2$ emissions
must be considered when routing flights for contrail avoidance (Teoh et al., 2022a). Also, climate optimized flight routing
might not always be feasible due to air traffic management considerations. As contrail cirrus induced climate forcing is related
to contrail ice particle numbers, another approach could be the reduction of contrail ice nucleating particles such as engine's
soot emissions (Kärcher and Yu, 2009; Schumann et al., 2013; Unterstrasser and Görsch, 2014; Moore et al., 2017; Bier et
al., 2017; Burkhardt et al., 2018; Voigt et al., 2021; Bier and Burkhardt, 2022). Significant reductions in soot emission and
corresponding ice particle number concentrations have been reported from previous in situ flight campaigns for the burning
of sustainable aviation fuel (SAF) blends due to their different fuel composition compared to conventional jet fuel (Moore



et al., 2017; Voigt et al., 2021; Bräuer et al., 2021b, a). Aside from aliphatic hydrocarbon chains, fuels from fossil feedstock
are naturally rich in cyclic aromatic hydrocarbons, polycyclic naphthalene and sulfur, which are typically quasi absent from all
SPK-type SAFs such as HEFA-SPK (Hydro-processed Esters and Fatty Acids - Synthetic Paraffinic Kerosene) derived from
sustainable feedstocks (Wilbrand, 2017; Moore et al., 2015). Cyclic and polycyclic hydrocarbons have been associated with
higher rates of incomplete combustion and subsequent soot formation compared to aliphatic chains (Cain et al., 2013; Brem
et al., 2015; Schripp et al., 2022). As a drop-in fuel, SAF blends (up to 50% blending ratio) require no modifications to aircraft
and airport infrastructure and are therefore readily usable and already used by airlines today (Rye et al., 2010; Pechstein, 2017),
leading to a reduction of contrail radiative forcing compared to the use of conventional jet fuel (Moore et al., 2017; Burkhardt
et al., 2018; Bock and Burkhardt, 2019; Voigt et al., 2021; Teoh et al., 2022b; Bier and Burkhardt, 2022). Additionally, SAF
fuels may provide a $CO_2$ reduction potential depending on the type of feedstock, its cultivation method and location, as well
as processing and distribution method (Buchspies and Kaltschmitt, 2017; Plassmann, 2017).

Up until now, in situ observations of contrails resulting from the burning of SAF have only been reported for SAF blended with
conventional jet fuel, leaving open questions on the effect of burning 100% SAF on contrail ice particle number concentrations
and microphysics. In this work, we present for the first time in situ observations of contrails from the combustion of 100% SAF
fully consisting of HEFA-SPK compared to conventional Jet A-1 fuel. These observations were made at cruise altitudes during
a measurement flight of the ECLIF3 campaign in a cooperation between DLR, Airbus, Rolls-Royce and Neste, where the DLR
Dassault Falcon 20-E5 was able to chase an Airbus A350-941 equipped with Rolls-Royce Trent XWB-84 engines and burning
100% HEFA-SPK fuel. We aim to contribute to the understanding of the soot and ice particle number reduction potential with
the use of 100% HEFA-SPK fuel compared to Jet A-1. We use these results to model the impact of the soot and ice particle
number reductions on the radiative forcing of contrails as well as to compare them to predictions of contrail ice particle number
concentrations made by the CoCiP model. By comparing these results to previous in situ contrail measurements of SAF blends
and a reference Jet A-1, the influence of the used engine and the fuel parameters can be qualitatively assessed. Further, size
distributions of ice particles in the investigated contrails provide insight into microphysical variations in the vertical contrail
distribution.

## 2 Experiment and Instrumentation

### 2.1 ECLIF3 campaign

Previous campaigns on Emission and CLimate Impact of alternative Fuels (ECLIF1) (Kleine et al., 2018) in 2015 and ECLIF2/-
NDMAX (Voigt et al., 2021; Bräuer et al., 2021b, a; Schripp et al., 2022) in 2018 focused on effects resulting from the use of
SAF blends with conventional jet fuel. The successor flight experiment campaign ECLIF3 for the first time allowed the inves-
tigation of effects from burning 100% HEFA-SPK on trace gas and particle emissions as well as contrail ice formation from a
large commercial passenger aircraft equipped with modern engines. The ECLIF3 campaign took place over two separate time
periods with three flights in April 2021 (ECLIF3-1) over the Mediterranean Sea and southern France and with six flights in
November 2021 (ECLIF3-2) off the Atlantic coast of France. Here, we focus on results from contrail flights during ECLIF3-1



as most of the flights in ECLIF3-2 were performed in ice-subsaturated conditions either designed as pure emission flights or in sublimating contrails, for which a quantitative data analysis of ice particles remains challenging. The first ever built Airbus A350-941 (registration: F-WXWB) equipped with two Rolls-Royce Trent XWB-84 engines and operated by Airbus served as the emission source aircraft. Fuel was supplied selectively from the two wing fuel tanks containing a reference fuel and 100% HEFA-SPK, respectively. This allowed for in-flight switching between two fuels for each engine and the operation of both engines on a single fuel using cross-feed valves. The DLR Dassault Falcon 20-E5 research aircraft followed the emission source aircraft to measure exhaust gases, volatile and non-volatile aerosol particles, and contrail ice particles. Two distance regimes were examined, with near-field measurements probing emissions at sub-second to few seconds age and far-field measurements probing contrails aged one minute to several minutes. This paper will address the contrail measurements during ECLIF3-1.

**Table 1.** Properties of fuels burned during the measurement flight. Fuel samples were taken from the tank before the measurement flight and an additional HEFA-SPK sample was taken from the fuel container for hydrogen content determination (ASTM D3701).

|  | Jet A-1 | HEFA-SPK |
| --- | --- | --- |
| Fuel composition | 100% | 100% |
| Aromatics (vol%) (ASTM D6379)* | 13.4 | 0.41* |
| Naphthalenes (vol%) (ASTM D1840)* | 0.35 | 0.002* |
| Hydrogen content (mass%) (ASTM D3701) | 14.08 | 15.11 |
| Carbon content (mass%)** | 85.90 | 84.89 |
| H:C mole fraction ratio | 1.95 | 2.12 |
| $EI_{CO_2}$ (g/kg) | 3149 | 3111 |
| Sulfur Total (mass%) (ASTM D5453) | 0.0211 | 0.0007 |

\* Aromatics (Naphthalene) content of HEFA-SPK is given in mass% and determined by GCxGC measurements due to the contents being below the ASTM D6379 (D1840) detection limits.

\*\*Carbon content based on the difference between 100% and hydrogen and sulfur content.

Two different jet fuels were investigated during the ECLIF3 flight campaign: A conventional reference Jet A-1 fuel supplied by TotalEnergies to Toulouse Blagnac airport and a 100% HEFA-SPK SAF produced by Neste Corporation. The fuel properties are described in Table 1 and show that the HEFA-SPK after logistics and at the point of use is nearly free of aromatics compared to the used Jet A-1 fuel which itself has a lower aromatics content than the global mean (Hadaller and Johnson, 2006). The Naphthalene content in the HEFA-SPK lies below the ASTM D1840 detection limit, while total sulfur content is higher by a factor of approximately 30 in the Jet A-1 but still 15 times below the maximum permitted value of the mass fraction of 0.3% (ASTM D5453). Further, the increased hydrogen content of HEFA-SPK compared to the used Jet A-1 results in a 1.2% reduced $CO_2$ emission index ($EI_{CO_2}$) for the HEFA-SPK fuel.

From the flights performed during the ECLIF3 campaign, the flight on 16 April 2021 stands out as having produced contrail data in ice-supersaturated conditions in a narrow range of atmospheric ambient and engine conditions as well as contrail ages.



During this flight over the Mediterranean Sea, race track patterns were flown in north–south direction west of Corsica and Sardinia while burning 100% Jet A-1 and HEFA-SPK fuels sequentially in both engines. The A350 was cruising at flight level FL350 $\approx$ 10.65 km while contrail ice particles formed from the engine exhaust. Contrails in the wake vortex phase were subsequently probed in the range of 48 m below to 96 m above the emission altitude in the secondary wake by the ice crystal instrumentation on board the Falcon.

## 2.2 Instrumentation and data evaluation

The DLR Falcon research aircraft is a Dassault Falcon 20E5 twin-engine business jet class aircraft, which has been highly modified to accommodate a multitude of atmospheric measurements. The airframe is built sufficiently robust to enable measurements in the high-turbulence regime behind large passenger aircraft. An extensive onboard instrumentation provides highly resolved aircraft position and dynamics data, as well as data on ambient meteorological conditions (Bögel and Baumann, 1991; Krautstrunk and Giez, 2012; Giez et al., 2017, 2020, 2021, 2023). These meteorological parameters such as temperature and pressure serve as input values for several calculated quantities discussed in this work. Horizontal aircraft distances were determined as the great circle distances based on WGS84 GPS latitude and longitude of both aircraft while vertical distances were determined as the difference between the WGS84 GPS altitudes of the two aircraft, resulting in a triangulated total distance. Using wind field measurements onboard the preceding Airbus A350 and the DLR Falcon ground speed, the time between plume emission and detection of the drifted contrail was iteratively calculated, thereby providing contrail ages at every point in time during the flight. This age is used to assign the correct aircraft and engine conditions at time of emission to the time of detection for the respective contrail. During the flights, underwing-mounted cloud probes and a cabin instrumentation for aerosol and trace gas measurements was employed and will be described in the following.

### 2.2.1 Ice particle and aerosol measurements

Ice particle data was obtained using the two instruments Cloud and Aerosol Spectrometer (CAS) (Baumgardner et al., 2001) and Cloud, Aerosol, and Precipitation Spectrometer (CAPS), containing a CAS-part and equipped with an additional Cloud Imaging Probe (CIP). The CAS-DPOL was mounted on the port side inner underwing position while the CAPS was mounted on the starboard inner underwing position resulting in a horizontal distance of approximately 7 m between the two instruments. Nominally, the CAS-parts of both probes cover a particle size range of 0.5 – 50 µm while the CIP covers 15 – 900 µm (Moser et al., 2023; Castro et al., 2023). Contrail ice particles at the probed age of one minute to several minutes typically lie in the size range covered by the forward scattering cloud probes (Voigt et al., 2010, 2011, 2017; Bräuer et al., 2021b), therefore we direct the focus to the analysis of CAS data. The CAS instruments operate with a $\lambda = 658$ nm laser where the detection of forward-scattered laser light delivers information on the number concentration and size distribution of ice particles. The CAS-part of the CAPS used in this experiment has been further developed compared to the single CAS, transitioning from a three-gain stage configuration to only two-gain stages with a variable gain stage overlap region. The CAS was calibrated using the methods of Rosenberg et al. (2012) and a size calibration conducted by the manufacturer Droplet Measurement Technologies Inc. was used for the CAPS. This calibration leads to an instrument specific size range of 0.66 – 41 µm for the CAS and a range of 0.5



140   – 51 μm for the CAPS. The CAS sampling area $\mathrm{SA_{CAS}}$ of $0.22 \pm 0.04$ mm$^2$ was mapped using a mono-dispersed stream of water droplets and the CAPS sample area $\mathrm{SA_{CAPS}}$ of $0.25 \pm 0.05$ mm$^2$ was provided by the manufacturer.

The size-dependent scattering cross section and subsequent size allocation of measured particles is influenced by their aspect ratio. Measurements by Gayet et al. (2012) show that contrail ice crystal shapes transition from quasi-spherical to more as-pherical within the first five minutes of contrail age (Jeßberger et al., 2013). Sanz-Morère et al. (2020) suggest that the change

in particle shape affects the radiative forcing of contrails. Therefore, for this analysis, an aspect ratio of 0.75 has been chosen with an ice particle refractive index of 1.31 and scattering cross sections derived from T-matrix simulations (Borrmann et al., 2000).

Aircraft engines emit a multitude of particle types called non-volatile and volatile particles according to the measurement

method used (see below). The non-volatile particle fraction in the exhaust primarily consists of soot from combustion, possibly containing sulfur or other volatile compounds that condense onto the soot particles in the cooling plume. Volatile particles include aqueous particles of sulfur compounds, unburned hydrocarbons, nitrates and others which are nucleated from pre-cursor gases in the rapidly cooling exhaust at engine exit. Depending on engine design, another source of volatile particles can be oil venting from the engine lubrication oil system (Timko et al., 2010; Fushimi et al., 2019; Ungeheuer et al., 2022).

The DLR-Falcon aerosol measurement system measures particle number concentrations using a set of butanol-based conden-sation particle counters (CPC), which have been customized for aircraft use, sampling ambient air through a forward-facing quasi-isokinetic inlet on top of the fuselage. To discriminate between total aerosol (i.e., non-volatile plus volatile particles) and non-volatile particles, three of the CPC are connected to a section of inlet line heated to 250°C (thermodenuder) to evaporate volatile components of the bulk aerosol. In addition, the CPCs sample with different lower cutoff-diameters $D_{50}$ using diffu-

sion screen separators (Feldpausch et al., 2006). At altitude, the lower cutoffs for non-volatile Particulate Matter (nvPM) are approximately 14 nm, 35 nm and 90 nm. Total particles are sampled down to 5 nm, not considering inlet losses. The particle counters are operated roughly at ambient pressure. Butanol-based CPCs encounter a loss of counting efficiency at low sample pressures (Noone and Hansson, 1990), which has been characterized in lab experiments with a setup similar to Hermann and Wiedensohler (2001) using size-selected Ag aerosol at 35nm from a tube furnace as reference aerosol. Based on this char-

acterization, a parameterized correction function is used to correct each particle counter individually. Due to ice activation of non-volatile particles, particles entering the aerosol inlet are significantly larger than the initial nvPM ice nuclei. An enrichment of aerosol particles occurs in the fuselage-top mounted inlet owing to slightly sub-isokinetic conditions resulting from true air speeds exceeding values that ensure isokinetic conditions. Based on a typical contrail ice particle size distribution, a correction factor to this enrichment of 76% was calculated according to the equations in Krämer and Afchine (2004) and Hinds (2012).

Assuming a nucleation efficiency of 96% according to Kärcher et al. (2018), this correction factor is applied to 96% of the median nvPM EI value from far-field contrail conditions. The overall uncertainty of particle counting during far-field measure-ments including uncertainties due to the size distribution dependent enrichment of aerosol particles in the inlet is conservatively estimated to 18%.



### 2.2.2 Trace gas and water vapor measurements

Total reactive nitrogen ($NO_y$), the sum of all reactive nitrogen species in the atmosphere, was measured using a well-established technique. On the surface of a heated gold tube, $NO_y$ species are reduced catalytically to NO followed by detection by a chemiluminescence detector (Ziereis et al. (2022), and references herein). The single channel instrument (a modified CLD 780TR, Eco Physics) worked with at 1 Hz sampling rate and received ambient air via an upper fuselage mounted inlet as well. The mean measurement uncertainty for the investigated contrail sequences was approximately 1 ppb.

Water vapor measurements were conducted with the water vapor mass spectrometer AIMS (Kaufmann et al., 2016, 2018), also receiving ambient air through an inlet mounted on the upper fuselage and designed for high-frequency measurements in the upper troposphere and lower stratosphere. The measured ambient water vapor mixing ratio together with ambient temperature and pressure measurements from the Falcon instrumentation is used to calculate the relative humidity with respect to ice. The uncertainty of 0.5 K in the temperature measurements and of 8-12% in the water vapor data combine to an uncertainty of around 15% for the RHi determination (the uncertainty of static pressure is negligible for the RHi calculation).

### 2.2.3 Determination of the apparent ice emission index

In order to be able to compare measurements at different contrail ages, stages of contrail vortex dynamics and position in the contrail, a dilution-corrected metric is needed. Generally, for particles of type X, an emission index is defined by (Beyersdorf et al., 2014):

$$\mathrm{EI_x} = \left(\frac{\Delta X}{\Delta \mathrm{Tr}}\right) \cdot \left(\frac{\mathrm{M_{air}}}{\mathrm{M_{tracer} \cdot \rho_{air}}}\right) \cdot \mathrm{EI_{tracer}}. \tag{1}$$

Here, $\Delta X$ and $\Delta \mathrm{Tr}$ are the enhancement of particle number and the mixing ratio of a gaseous tracer, which is assumed to have the same mixing behavior as the particles above their respective background. $\mathrm{M_{air}}$ and $\mathrm{M_{tracer}}$ are the molar masses of air and the tracer gas, $\rho_{air}$ is the density of ambient air and $\mathrm{EI_{tracer}}$ is the emission index of the used tracer gas.

Analogous to the emission index for particle number concentrations, the apparent contrail ice emission index (AEI) describes the number of contrail ice particles formed from aircraft exhaust per kg of fuel burned. By using a dilution tracer and measuring in the secondary wake, a dilution-corrected metric is achieved, which can be used to draw conclusions for the two probed fuels independent of contrail age. The average enhancement of $CO_2$ in contrails during the measurements discussed here was in the range of the background variability of the long-lived greenhouse gas $CO_2$. In contrast, due to the relatively low background concentration and variation in the atmosphere at cruise altitudes, reactive nitrogen species $NO_y$ are a useful tracer as an indication of contrail dilution. At high engine power settings, emitted nitrogen compounds mainly consist of NO (>80%) and $NO_2$, while NO rapidly reacts with $O_3$ to form $NO_2$ (Schulte et al., 1997). Therefore, in order to minimize induced uncertainties due to $CO_2$ background variability, $NO_y$ was chosen as dilution tracer. Measured $NO_y$ contains all nitrogen species processed in the atmosphere from the initial $NO_x$ emissions. As stated in Schulte et al. (1997), the $NO_x$ emission index is defined in mass units of $NO_2$. Therefore, $NO_2$ molar mass is used for the $NO_y$ tracer together with the molar mass of air and the gas constant, resulting in the following equation dependent on measured parameters:





$$ \mathrm{AEI}/\mathrm{kg}^{-1} = 1.807 \cdot 10^{12}\,\mathrm{J\,kg^{-1}\,K^{-1}} \cdot \left( \frac{\Delta N/\mathrm{cm}^{-3}}{\Delta \mathrm{NO_y}/\mathrm{ppbv}} \right) \cdot \left( \frac{T_\mathrm{amb}/\mathrm{K}}{p_\mathrm{amb}/\mathrm{hPa}} \right) \cdot \mathrm{EI_{NO_x}}/\mathrm{g\,kg^{-1}}. \tag{2} $$

During a chosen contrail measurement interval, $\Delta N$ and $\Delta \mathrm{NO_y}$ are the time integrals over ice particle number concentration and $\mathrm{NO_y}$ concentration enhancements over their respective atmospheric backgrounds and $T_\mathrm{amb}$ and $p_\mathrm{amb}$ are the mean ambient temperature and pressure. $\mathrm{EI_{NO_x}}$ refers to the combustion state dependent $\mathrm{NO_x}$ emission index. $\mathrm{EI_{NO_x}}$ predictions based on the Rolls-Royce prediction method (P3T3 method) and performed by Rolls-Royce are used for the compressor outlet temperature (T30) ranges corresponding to the far-field ice particle measurements presented here. Resulting $\mathrm{EI_{NO_x}}$ values are 17.4 g/kg for the T30 values of Jet A-1 and 17.5 g/kg for the 6.5 K higher T30 values during combustion of HEFA-SPK.

Time intervals are defined to begin when ice particle number concentrations start significantly departing from background levels and to end when they return to the background. Fluctuations in background concentrations for ice particles are negligible compared to ice particle enhancements in contrails and are therefore assumed to be zero. Background levels of $\mathrm{NO_y}$ are determined for every time interval and lie at a mean value of $(0.76 \pm 0.12)$ ppb for our contrail sequences (where ppb denotes molar mixing ratio in nmol/mol).

If the Falcon is fully immersed in the contrail, a quasi-homogeneous spatial distribution of ice particles at every point of the aircraft is assumed. In certain measurement sequences at the edge of contrails, this assumption does not hold and the ice particle instrumentation may be more immersed in the contrail than the trace gas inlet or vice versa due to the distance between the two instruments. This situation leads to a strong discrepancy in concentration measurements between ice particles and trace gas, resulting in disagreeing time-resolved peak shapes between the two quantities. This indication is used to mark invalid sequences and to exclude them from further investigation as they would produce misleading apparent ice emission indices based on respective dilutions not equal to those in the ice particle measurement.

### 2.2.4 Uncertainties of apparent ice emission indices

In the following section, single contributions to the total uncertainty of apparent ice emission indices are explored and the total uncertainty is quantified. Following Eq. 2 for the definition of AEI, uncertainties of individual parameters are taken into account: (a) ice particle measurements $(\partial \Delta N_\mathrm{meas})$, (b) $\mathrm{NO_y}$ measurement $(\partial \Delta \mathrm{NO_{y\,meas}})$, (c) $\mathrm{NO_y}$ background determination $(\partial \Delta \mathrm{NO_{y\,bckgr}})$, (d) ambient temperature measurements $(\partial \Delta T_\mathrm{amb}) = 0.5$ K, (e) ambient pressure measurements $(\partial \Delta P_\mathrm{amb}) = 0.5$ hPa, and (f) $\mathrm{EI_{NO_x}}$ prediction uncertainty $(\partial \Delta \mathrm{EI_{NO_x}}) = 10\%$.

Ice particle number concentrations according to Eq. A1 depend on several parameters with individual uncertainties. Uncertainty due to counting statistics is estimated by $(\partial n_i) = (\sum_{i=\mathrm{t_{begin}}}^{i=\mathrm{t_{end}}} n_i)^{-\frac{1}{2}}$, where $\sum_{i=\mathrm{t_{begin}}}^{i=\mathrm{t_{end}}} n_i$ is the sum over 1 Hz counts from beginning to end of a single contrail time interval. Further, sample area uncertainty has been determined as approximately $(\partial \mathrm{SA_{CAS}}) = 18\%$ for the CAS-DPOL and $(\partial \mathrm{SA_{CAPS}}) = 20\%$ for the CAPS via sample area calibrations and the uncertainty in using true air speed (TAS) as sample air speed (SAS) has been determined as $(\partial \mathrm{TAS}) = 7\%$ in simulations. These uncertainties are propagated using Gaussian error propagation, resulting in a total ice particle measurement uncertainty $(\partial \Delta N_\mathrm{meas})$.





$NO_y$ measurements are affected by uncertainties consisting of the instrument and measurement accuracy $\partial NO_{y\,acc}$ and the uncertainty determined by atmospheric background fluctuations $\partial NO_{y\,bckgr}$. The total uncertainty of AEI is finally calculated by using the Gaussian error propagation described in Eq. 3:

$$\partial AEI = \pm \sqrt{\begin{array}{l}\left(\dfrac{\partial AEI}{\partial \Delta N}\partial \Delta N_{meas}\right)^2 + \left(\dfrac{\partial AEI}{\partial \Delta NO_y}\partial \Delta NO_{y\,meas}\right)^2 + \left(\dfrac{\partial AEI}{\partial \Delta NO_y}\partial \Delta NO_{y\,bckgr}\right)^2 + \\ \left(\dfrac{\partial AEI}{\partial \Delta T_{amb}}\partial \Delta T_{amb}\right)^2 + \left(\dfrac{\partial AEI}{\partial \Delta P_{amb}}\partial \Delta P_{amb}\right)^2 + \left(\dfrac{\partial AEI}{\partial EI_{NO_x}}\partial \Delta EI_{NO_x}\right)^2\end{array}} \tag{3}$$

This amounts to a median AEI uncertainty of approximately 33% for the determination of absolute AEI values. When only considering non-systematic errors, which do not influence the relative difference of values to each other, the median AEI uncertainty is reduced to approximately 26%.

### 2.2.5 The CoCiP model

CoCiP is a Lagrangian model that traces individual contrail segments forming along given flight routes (Schumann, 2012). For the present application, CoCiP prescribes the ambient atmosphere using numerical weather analysis data of the Integral Forecasting System (IFS) of the European Center for Medium-Range Weather Forecasts (ECMWF) with 0.125° spatial and 1 h temporal resolution, on model levels with about 300 m vertical resolution at the given levels. The model simulates the contrail as forming behind the Airbus aircraft for given flight route, speed, fuel consumption, and fuel hydrogen content, resulting in a soot emission index computed for the given engine type as in Teoh et al. (2022a). In this application, CoCiP uses the Falcon-measured relative humidity changed by the difference in relative humidity in the IFS values at the contrail position and at the Falcon position (RHi at Falcon position as measured plus RHi at contrail position as derived from IFS data minus RHi at Falcon position as derived from IFS data). The Schmidt-Appleman threshold temperature is computed using the actual overall propulsion efficiency $\eta$ (Schumann, 1996; Poll and Schumann, 2021). The model assumes that the soot particles emitted into the young exhaust plumes act as condensation nuclei for ice particle formation when humidity exceeds liquid saturation locally in the engine exhaust while mixing with ambient cold air. The resultant droplets freeze soon thereafter because of low temperature. The initial contrail phase is prescribed in terms of the initial number of ice particles per flight distance, the initial geometrical plume cross-section area, and the initial ice water content. The initial ice water content is computed from $EI_{H_2O}$ times fuel flow per distance plus the amount of humidity entrained from ambient air into the contrail, assuming uniform mixing across the plume cross-section, the area of which varies with age. The initial number of ice particles per flight distance is set equal to the number of soot particles emitted per flight distance into the contrail. Part of the ice particles sublimates in the sinking wake vortex when the ambient air is below ice saturation. This is modelled in CoCiP thermodynamically as a function of adiabatic warming due to contrail sinking. Further losses occur due to detrainment from the wake by turbulent mixing with dry ambient air. Contrails spread vertically mainly by turbulent mixing excited by shear and limited by stable stratification. The further changes in contrail bulk ice physics are approximated as a function of ice water content and ice particle number $N_{ice}$ per flight distance assuming saturation inside the contrail. The mean local ice particle concentration $n_{ice}$ is computed from





the number of ice particles per flight distance divided by the plume cross-section. The number of ice particles per unit plume length decreases slowly due to parameterized aggregation between contrail particles and turbulent mixing losses. The contrail ice water content grows by uptake of ambient humidity entering the plume when mixing with ambient ice supersaturated air. When mixing with subsaturated air, the ice water content decreases and the contrail disappears after some time when the contrail ice water content reaches zero. Hence, as observed (Li et al., 2023), the contrails persist some time in ice-subsaturated air. The volume mean particle radius $r_{vol}$ is computed from the ice mass and particle number per volume. The volume-mean ice particle size is used to compute the mean fall speed (Spichtinger and Gierens, 2009). The vertical motion of the contrail follows the sum of ambient vertical velocity and fall speed. Sedimentation also contributes to enhanced vertical widening of the plume cross-section. Contrails terminate when all ice water content is sublimated by mixing with dry air or when the ice particles fall below the model bottom. CoCiP has been compared extensively with in situ and remote sensing observations (Voigt et al., 2010; Jeßberger et al., 2013; Schumann and Graf, 2013; Schumann et al., 2013, 2015, 2017; Voigt et al., 2017, 2022; Schumann et al., 2021).

### 2.2.6 Estimating contrail cirrus radiative forcing within a climate model

In order to estimate the climate impact of contrail cirrus, we conducted two global climate simulations, one for a global aircraft fleet using Jet A-1 fuel and the other using 100% HEFA-SPK. The simulations were performed with the general circulation model ECHAM5 (Roeckner et al., 2003) coupled with the Hamburg Aerosol Module (Stier et al., 2005). Within this model, a contrail cirrus module, CCMod, was implemented, consisting of parameterizations of the processes important for the lifecycle of contrail cirrus and their microphysical properties. The CCMod parameterization introduces a separate cloud class, contrail cirrus, consistent with the model's cloud scheme, which interacts with natural cirrus clouds (Burkhardt and Kärcher, 2009). Water and heat budgets are closed within ECHAM5-CCMod which means that contrails and cirrus compete for available water vapor and that diabatic heating due to microphysical and radiative processes within contrails has an impact on the atmospheric state. The model uses a microphysical two-moment scheme (Lohmann et al., 2008; Bock and Burkhardt, 2016a). Contrail formation follows the Schmidt-Appleman criterion (Schumann, 1996). The ice nucleation parameterization, based on Kärcher et al. (2015), estimates the number of soot and ambient particles that form droplets, which subsequently freeze. The number of droplets forming and subsequently freezing is not only dependent on the number, size and hygroscopicity of soot and ambient atmospheric particles but crucially also on the atmospheric state with fewer droplets forming when contrail formation happens close to the Schmidt-Appleman criterion. The ice crystal loss in the vortex phase is parameterized according to Unterstrasser (2016). The parameterization is based on large eddy simulations (LES) with detailed microphysics and Lagrangian ice crystal tracking. The evolution of young contrails is simulated and is dominated by the interplay of microphysics and wake vortex dynamics while ice crystal loss is dependent on atmospheric and aircraft related parameters. The number of ice crystals that form behind an aircraft and survive the vortex phase has been shown to be a predictor of contrail cirrus radiative forcing (Burkhardt et al., 2018; Bier and Burkhardt, 2022). Bier and Burkhardt (2022) show that contrail ice nucleation is in good agreement with in situ measurements and that there is a large spatial variability of ice nucleation for fixed soot number emissions with nucleation rates much smaller at lower altitudes and in tropical areas. Contrails are initialized within the model





calculating a consistent set of ice crystal number concentration and contrail cross sectional area from the apparent ice number emission index resulting from the ice nucleation and ice crystal survival parameterizations. Ice water content at initialization

results from the water emissions and to a larger degree from the environmental ice supersaturation. Contrails persist and spread within the fraction of the grid box that is ice supersaturated (Burkhardt et al., 2008). The nucleated ice crystals are assumed to be homogeneously distributed within the plume volume which grows initially due to turbulent diffusion and later due to the combined action of vertical wind shear, increasing the contrail vertical tilt, and differential sedimentation of the contrail ice crystals (Bock and Burkhardt, 2016a). While contrail ice crystals initially grow by efficiently relaxing high plume ice

supersaturations, in the later life cycle ice crystal number concentrations are often not large enough to relax atmospheric ice supersaturation within a time step. Therefore, we limit ice crystal growth for low ice crystal number concentration (Bock and Burkhardt, 2016a). Otherwise ice microphysical processes within the later life cycle of contrails are the same as in natural ice clouds and are therefore calculated by the model's microphysical scheme. Ice crystal number concentration, ice water content and optical depth of young contrails and contrail cirrus were shown to be in good agreement with in situ and satellite based

observations (Bock and Burkhardt, 2016b).

Air traffic (flight distance and fuel consumption) is prescribed by the AEDT flight inventory (Wilkerson et al., 2010) for the year 2006. For the reference simulation, representing the average fleet currently in service, we used a soot emissions index of $1.0 \times 10^{15}$ kg$^{-1}$ (Table 2), which is described in Teoh et al. (2023b) as the global fleet average for the year 2019. In the second simulation, portraying the use of 100% SAF for the entire fleet, we reduced the soot number emission index consis-

tent with the measurements of the apparent ice number emission index (Section 3.2). In the soot rich regime for temperatures several degrees below the Schmidt-Appleman contrail formation temperature, the decrease in ice crystal numbers is equal to the decrease in emitted soot numbers. We therefore prescribe a reduction in soot number emissions of 60% resulting in a soot number emission index of $4.0 \times 10^{14}$ kg$^{-1}$. As will be discussed below, this reduction in ice crystal numbers may be partly due to a reduction in soot number emissions and partly due to the low sulfur content of the fuel. The $H_2O$ emission index and

the combustion heat for kerosene were set consistent with measurements from the ECLIF1 measurement campaign while the respective values for 100% SAF were set to values from HEFA-SPK available during measurement campaigns in November 2021. The overall propulsion efficiency is estimated for the current high bypass ratio fleet (Epstein, 2014). The simulation parameters are summarized in Table 2. The spatial resolution of the climate model was set to T63L41 (equivalent to ~200 km grid boxes at the equator and a vertical resolution of approximately $500$ m at the tropopause) with a corresponding timestep

of 10 min. We perform our simulations using an air traffic inventory of the year 2006 and scale our results to the year 2018 as done in Lee et al. (2021).

## 3 Results

### 3.1 Measurement of contrails at cruise altitude

This section presents an in-depth analysis of in situ contrail ice particle measurements at cruise conditions during the contrail

flight on 16 April 2021 behind an Airbus A350-941 equipped with Rolls-Royce Trent XWB-84 engines and burning 100%




**Table 2.** Input parameters for the reference fuel and the SAF fuel global climate simulations

| Parameter | Reference Run | SAF Run |
|---|---|---|
| $EI_{soot}$ [$10^{15}$/ kg-fuel] | 1.0 | 0.4 |
| $EI_{H_2O}$ [kg/ kg-fuel] | 1.24 | 1.37 |
| Combustion heat [$10^6$ J /kg-fuel] | 43.2 | 44.1 |
| Overall propulsion efficiency | 0.36 | 0.36 |

reference Jet A-1 and $100\%$ HEFA-SPK successively. One other flight from ECLIF3-1 focused on emission measurements and the other contrail flight did not have the option to provide 100% HEFA-SPK to the two engines, as this was the first test flight. Far-field contrail measurements (distance between 19 km and 35 km) of Jet A-1 fuel were conducted between 11:18:22 UTC and 11:33:00 UTC while contrails from burning HEFA-SPK were measured between 11:42:16 UTC and 11:52:55 UTC.

A detailed time series plot of this measurement sequence is shown in Figure 1 where the sequence of Jet A-1 and HEFA-SPK are shaded in gray and green respectively. Contrail encounters are marked by the strong increases in ice crystal numbers and soot numbers in line with enhanced $CO_2$ and $NO_y$ measurements. The contrail measurements are clearly identified in the measurement time series.

The difficulty of drawing any reliable conclusions from time series alone illustrates the need for the dilution-corrected ap-
parent contrail ice emission index. In the time series, groups of peaks can be seen where each individual peak of ice particle measurements constitutes a contrail encounter with a peak being defined as the time period between the departure from and return to ice particle background concentrations. However, not all of the collected data shown in Figure 1 can be used to calculate valid apparent ice emission indices. The selection of data sequences to derive apparent ice emission indices is described in Appendix B.

After filtering data according to the above described criteria, a data set remains with quality-controlled valid data points, which can be used to calculate comparable AEI values. Table 3 lists the ranges, means and standard deviations as computed for the reduced data set. Here, in addition to the columns for Jet A-1 and HEFA-SPK, a third column shows the values for HEFA-SPK when applying a filter to the data set that only accepts contrail encounters where the time series of ice particle measurements and $NO_y$ correlate better than $60\%$. This way, aforementioned sequences with preferential contrail immersion of either the
aerosol and trace gas inlets or the ice particle instrumentation are quantified and can be disregarded for further analysis. Jet A-1 data is coincidentally not influenced by this filter and therefore no separate column is given.

For Jet A-1, measured contrails were between 104 s and 142 s old while HEFA-SPK contrails were slightly younger between 73 s and 92 s. While performing far-field measurements in contrails, the distance between the source aircraft and the DLR Falcon continuously increases due to the limited propulsion performance of the Falcon compared to the Airbus A350, thereby
intentionally leading to the range of contrail ages for each fuel sequence. As the measurements were taken in ice supersaturated conditions within the vortex regime in the secondary wake of the contrail near flight altitudes, ice particle number concentra-



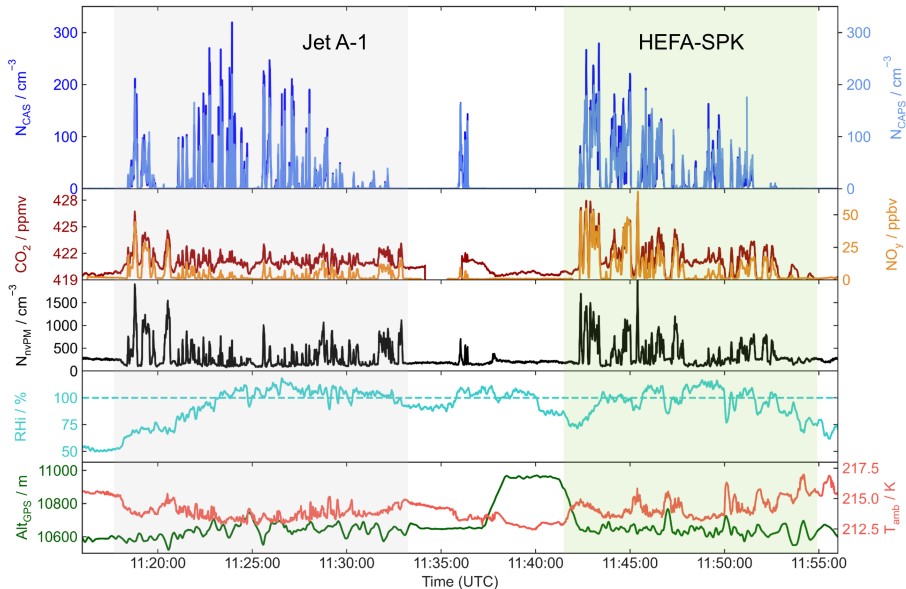

**Figure 1.** Time series of 1 Hz data during far-field measurement sequences of Jet A-1 (gray shading) and HEFA-SPK fuel (green shading) showing number concentrations of ice crystals larger than at least 0.5 µm measured by the CAS ($N_{CAS}$) and CAPS ($N_{CAPS}$) instruments, $CO_2$ and $NO_y$ mixing ratios, number concentrations of nonvolatile particles > 14 nm. ($N_{NvPM}$), relative humidity over ice (RHi) during the measurements with indicated saturation (dashed line), and the GPS altitude of Falcon ($Alt_{GPS}$) with the Falcon measured ambient temperature ($T_{amb}$).

**Table 3.** Contrail measurement conditions for the respective fuel measurement sequences. A ice-$NO_y$ time series correlation filter of 60% further reduces HEFA-SPK data (third column) while Jet A-1 data is not further reduced by this filter.

|  | Jet A-1 | HEFA-SPK | HEFA-SPK (correlation > 60%) |
|---|---|---|---|
| Contrail age (s) | 104–142 | 73–92 | 75–87 |
| Sampling time with/without filtering (s) | 183/750 | 123/522 | 55/522 |
| Ambient RHi (%) | 101–115 | 102–113 | 102–112 |
| Ambient T (K) | 213.3 (±0.2) | 213.8 (±0.3) | 213.7 (±0.3) |
| $\Delta T_{SA}$ (K) | -11.8 (±0.2) | -11.6 (±0.4) | -11.7 (±0.3) |
| Altitude of source aircraft (m) | 10626 (±4) | 10621 (±4) | 10622 (±2) |
| Speed of source aircraft (Mach) | 0.846 (±0.004) | 0.852 (±0.001) | 0.852(±0.001) |
| Fuel flow (kg/h) per engine | $FF_{Jet}$ (±0.6%) | $FF_{Jet}$ + 1.9% (±0.2%) | $FF_{Jet}$ + 1.8% (±0.1%) |
| Engine T30 (K) | $T30_{Jet}$ (±0.9) | $T30_{Jet}$ + 6.5 (±0.8) | $T30_{Jet}$ + 6.2 (±0.4) |

Measurement conditions filtered for RHi > 100%. Sampling time is sum of valid contrail encounters. Arithmetic standard deviation (± a.s.d.) given in brackets. The original sampling time before filters is given behind the slash.





tions and tracers are assumed to get diluted similarly so that the AEI are not expected to be affected by the age difference. The

total sampling time in all valid contrails shown in Table 3 has the largest discrepancy between the two fuels and affects the
statistical significance of data. As the data was filtered to only include ice supersaturated data points, all shown measurements
were taken at RHi $> 100\%$ with very similar ranges between Jet A-1 and HEFA-SPK. The temperature-related parameters
such as ambient temperature and the difference to Schmidt-Appleman threshold temperature ($\Delta T_{SA}$) were within each other's
standard deviations from the respective means. The emitting Airbus A350 was flying at nearly constant altitudes on flight level
FL350 at steady Mach numbers for the two fuel burning sequences. Engine combustion conditions are described by the pa-
rameters high pressure compressor outlet temperature T30 and the fuel flow FF. Their fluctuations are relatively small for each
individual fuel sequence where average fuel flow and T30 were respectively $1.9\%$ and $6.5$ K higher for HEFA-SPK burning
sequences compared to Jet A-1. The differences of ice-$NO_y$ correlation filtered HEFA-SPK data points and non-filtered data
points is negligible except for the sampling time, which reduces the overall statistics of HEFA-SPK measurements. However,
this step leads to an increased quality of data by quantifying spatial inhomogeneities and focusing on homogeneous contrail
encounters.

Overall, the quality-controlled and reduced data set with the conditions shown in Table 3 provides a solid basis for compar-
ison of AEI for HEFA-SPK fuel burning compared to Jet A-1, considering fluctuations of relevant atmospheric and aircraft
parameters are statistically distributed.

### 3.2   Impact of 100% HEFA-SPK on the apparent ice number emission index AEI

With the quality-controlled data set described in previous sections (filtered for $> 60\%$ ice-$NO_y$ correlation), it is possible to
compare AEI for Jet A-1 and HEFA-SPK. Figure 2 shows comparisons of AEI for Jet A-1 and HEFA-SPK (individual data
points, medians and arithmetic standard deviations) versus a set of parameters relevant in ice particle formation. Figure 2 (a)
relates contrail ice particle numbers to measured soot particle emission indices corresponding to the contrail measurement data
while further panels compare AEI depending on the fuel parameters (b) naphthalene, (c) aromatics, and (d) sulfur.

**Table 4.** AEI and $EI_{nvPM}$ for Jet A-1 and HEFA-SPK (based on 60% correlation filtered data) for the ECLIF3 flight on 16 April 2021

| Property | Jet A-1 ($\times 10^{14}$ kg$^{-1}$) | HEFA-SPK ($\times 10^{14}$ kg$^{-1}$) |
| --- | --- | --- |
| AEI median | $7.8 \pm 4.0$ | $3.4 \pm 1.5$ |
| AEI mean | $8.7 \pm 4.0$ | $3.9 \pm 1.5$ |
| nvPM EI median | $9.5 \pm 3.0$ | $6.1 \pm 0.7$ |
| nvPM EI mean | $10.3 \pm 3.0$ | $6.4 \pm 0.7$ |

Figure 2 (a) shows a reduction in median AEI for 100% HEFA-SPK compared to Jet A-1 of $56\%$. The absolute AEI
decreases from $7.8 \times 10^{14}$ kg$^{-1}$ to $3.4 \times 10^{14}$ kg$^{-1}$. At the same time, median nvPM EI are reduced by $35\%$ from $9.5 \times 10^{14}$



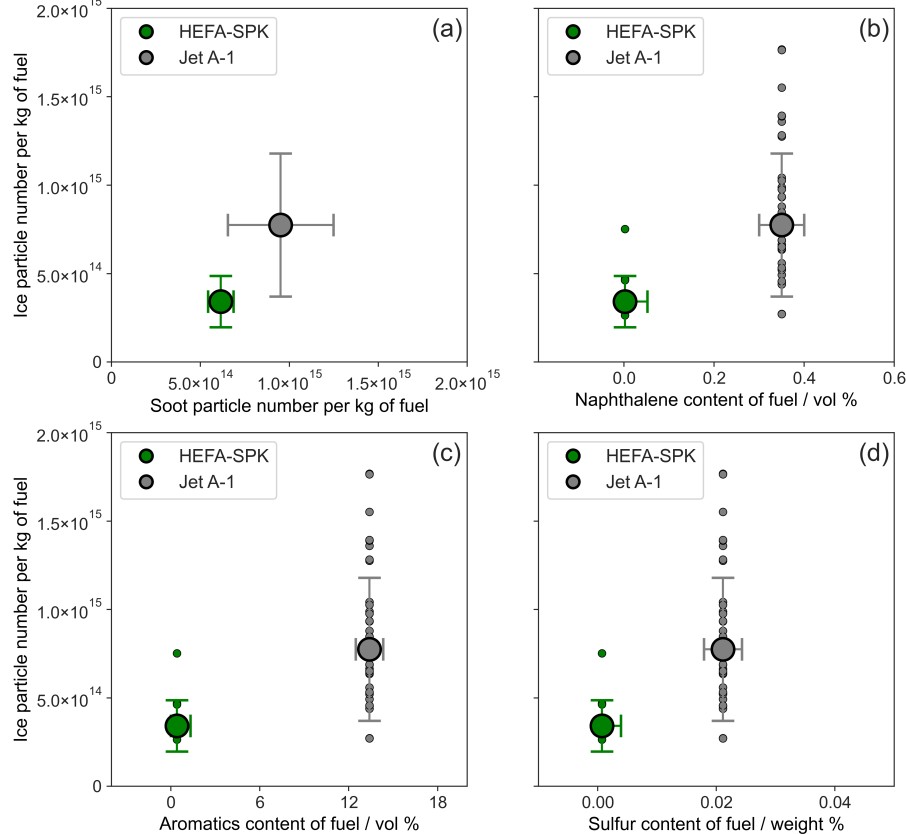

**Figure 2.** AEI of ECLIF3 (flight on 16 April 2021) contrail measurements of Jet A-1 and HEFA-SPK fuels versus (a) nvPM emission indices, and versus respective fuel parameters (b) naphthalene content, (c) aromatics content, and (d) sulfur content. x-axis values are shown in Table 1. Single scattered data points show data from individual contrail crossings, large circles represent the respective medians and the bars give the arithmetic standard deviation. Note: aromatics (naphthalene) content of HEFA-SPK is based on GCxGC measurements (%w/w) and not ASTM D6379 (D1840) due to being below the detection limit of these methods. x-axis error bars are the reproducibility of the respective ASTM fuel property detection method (also used as conservative proxy for GCxGC uncertainty).

kg$^{-1}$ to $6.1 \times 10^{14}$ kg$^{-1}$. Both fuels were probed at very similar conditions within a single flight, therefore it is reasonable to attribute these reductions to properties of the probed fuels, which are explored in the following. Due to the higher binding energy between atoms in aromatics compared to aliphatic compounds, incomplete combustion and subsequent soot formation is enhanced for these compounds (Cain et al., 2013; Brem et al., 2015; Schripp et al., 2022). Naphthalene as a polycylic aromatic compound is especially conducive to soot formation and a reduction in naphthalene has been experimentally demonstrated to reduce apparent contrail ice emission indices (Voigt et al., 2021; Bräuer et al., 2021b). As shown in Figure 2 (b) and (c), the naphthalene and aromatic contents are reduced to below their ASTM D1840 and ASTM D6379 detection limits for HEFA-SPK compared to Jet A-1. Hence the soot reduction of 35% can be explained by this reduction in aromatics and naphthalene in the




SAF. It is also worth mentioning that despite a strong or an almost complete reduction of aromatics and naphthalene, there is still a residual amount of soot particle emissions of $6 \times 10^{14}$ nvPM EI per kg of fuel that contribute to contrail formation from the combustion of HEFA-SPK. Also, the question arises why the $56\%$ reduction of ice particles is larger than the corresponding $35\%$ reduction in soot number emissions. The reduction of sulfur content of the HEFA-SPK from 0.0211 mass% to 0.0007 mass%, corresponding to a reduction of $97\%$ for HEFA-SPK compared to Jet A-1 shown in Figure 2 (d) could give one

possible explanation. Sulfur contained in fuels can result in emissions of gaseous $SO_2$, which can lead to the formation of very small sulfuric acid droplets (Petzold et al., 1997; Schumann et al., 2002; Jurkat et al., 2011; Kärcher, 2018). Moreover, models show that sulfur can activate the initially hydrophobic soot particles (Jones and Miake-Lye, 2023). Thereby it facilitates ice particle nucleation by increasing the hydrophilicity of the soot particles (Wong and Miake-Lye, 2010). Our results point towards a possibly reduced soot particle activation into ice particles due to the low fuel sulfur content and might explain

stronger reduction of ice particles than from the reduction of soot alone. A similar effect has been suggested by Jones and Miake-Lye (2023) to explain observations of reduced soot activation into ice for a low-sulfur HEFA-SPK blend measured during the ECLIF2 campaign (Voigt et al., 2021).

At the same time, a larger spread in AEI values is observed for the Jet A-1 data points compared to HEFA-SPK due to the larger sample size of Jet A-1 measurements taken at cruise conditions. A large range of ice crystal number concentrations has been

measured in young contrails (Heymsfield et al., 2010; Voigt et al., 2010, 2011, 2017; Schumann et al., 2013, 2017; Jeßberger et al., 2013; Gayet et al., 2012; Chauvigné et al., 2018) due to the strong dynamical variations in humidity and temperature in the expanding plume as well as dilution in the vortex phase. The entrainment of ambient air in the primary vortex and the secondary wake leads to a multitude of conditions within the contrail, which can lead to sublimation locally, reflected in variations in ice crystal number concentrations (Lewellen et al., 2014). This explains the observed variations and shows that the

assumption of AEI values not being influenced by vortex phase dynamics is not entirely true in reality. In addition, variations in altitude, position in the contrail, age of contrail and resulting state of development add to the distribution of AEI values due to ice crystal loss or measurement fluctuations as described in 2.2.3.

Medians and means of soot data are based on nvPM emission indices measured in far-field contrail measurements at the same time as the presented apparent ice emission indices. Although nvPM emissions are preferentially measured in ice-free near-

field emission conditions, data from far-field contrail sequences were chosen due to their sensitivity on the Mach number of the source aircraft and the lack of near-field data at similar Mach numbers as in the far-field ice measurements. However, nvPM emission indices are slightly higher than their corresponding AEI values, which indicates the presence of interstitial soot, resulting possibly from a reduction in nucleation efficiency for the low sulfur fuel, local sublimation effects and/or fluctuations in nucleation efficiency. Further, the correction factor accounting for particle enrichment in the aerosol inlet is

based on assumptions of ice particle size and nucleation efficiency and effects such as particle trajectory deviations due to streamline compression and expansion at the fuselage (Afchine et al., 2018) are not considered. This highlights the need for ice-free near-field emission measurements for detailed analyses of aerosol emissions. Although the nvPM EI values in the case presented in Figure 2 (a) are subject to the described sampling uncertainties, they nonetheless are a valuable indicator of soot





particle activation, which is assumed to be the dominant ice particle activation mechanism in the soot-rich regime (Kleine et al.,
2018; Kärcher, 2018).

### 3.3    Impact of fuel composition and engine type on the apparent ice number emission index AEI

These data can now be compared with results from the preceding ECLIF1 and ECLIF2/NDMAX campaigns where particle
emissions and apparent contrail ice particle emissions were investigated in a similar manner as in ECLIF3. It needs to be kept
in mind that we are not able to independently and systematically vary single parameters and investigate their isolated influence
on ice particle concentrations. This is amplified when comparing results from several campaigns where different engines, fuels
and measurement platforms were used. Aerosol inlet systems were not identical and we estimate a 20% uncertainty for the
intercomparison between campaigns for soot particle measurements. During the ECLIF1 and ECLIF2/NDMAX campaigns,
the blends of a Fischer-Tropsch based synthetic jet fuel with Jet A-1 (SSF1) and two blends of 30% and 50% biomass-based
HEFA-SPK alternative jet fuel with Jet A-1 (SAF2 and SAF1) (Schripp et al., 2022) were compared to a reference Jet A-1 fuel
(Ref2) as described in Voigt et al. (2021). However, a qualitative assessment of the influence of fuel composition and the type
of engine used can be achieved by comparing AEI from different campaigns against their respective soot emissions and fuel
constituents.

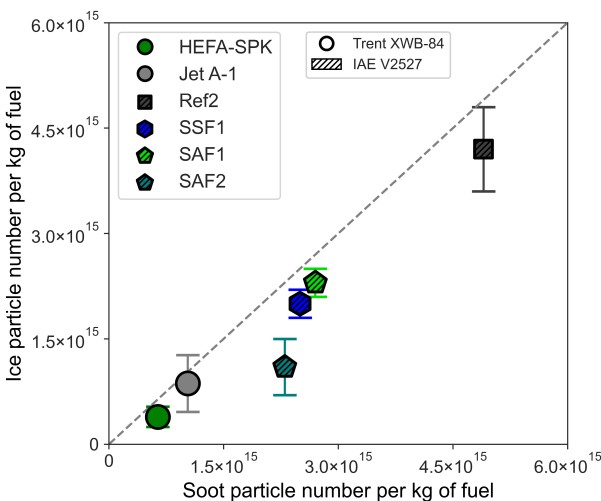

**Figure 3.** Apparent ice emission indices versus nvPM emission indices for Jet A-1 and HEFA-SPK fuel from the ECLIF3 campaign using
a Rolls-Royce Trent XWB-84 engine (circles) compared to fuels investigated during ECLIF1 and ECLIF2/NDMAX (Ref2, SSF1, SAF1,
SAF2) using an IAE V2527 engine (hatched symbols) (Voigt et al., 2021). The symbols represent means of the respective quantities in order
to facilitate comparability between ECLIF1, ECLIF2/NDMAX, and ECLIF3 data. The dashed line shows the ideal 1:1 relationship between
AEI and nvPM EI.





In Figure 3, mean apparent ice emission indices versus mean nvPM emission indices of these fuels are shown together with the mean AEI and nvPM EI of ECLIF3 as described in Table 4. We find a nearly linear relationship between ice particle numbers

and nvPM particle numbers for the different fuels and different engines. Measurements during ECLIF1 and ECLIF2/NDMAX were conducted behind the DLR A320 Advanced Technology Research Aircraft (ATRA) equipped with IAE V2527-A5 engines with higher soot emissions compared to the Rolls-Royce Trent XWB-84. Many relevant measurement conditions were similar for the ECLIF1 and ECLIF2/NDMAX data and the data was also filtered to only include data points with relative humidity over ice of $> 100\%$. The fuel/engine combination investigated during ECLIF3 have lower nvPM EI and therefore lower

AEI compared to the fuel/engine combination probed during ECLIF1&2. In particular the newer Rolls-Royce Trent XWB-84 engine exhibits lower soot particle emissions compared to the old IAE V2527 engine probed during ECLIF1&2. Also, the aircraft were different, with the smaller and lighter A320 ATRA chased previously and the A350-MSN1 probed during ECLIF3. It is especially interesting that soot and apparent ice particle emissions of the ECLIF3 Jet A-1 lie below the SAF blends from previous campaigns. In order to disentangle fuel and engine effects on the emissions, relevant fuel properties are compared in

Figure 4.

AEI values from the three ECLIF campaigns from both aircraft/engines are shown in Figure 4 plotted against the fuel parameters (a) naphthalene, (b) hydrogen content, (c) aromatics content, and (d) sulfur content. The stronger bonding of the mono- and polycyclic aromatic compounds explains their propensity to form soot precursors. An increase in hydrogen content thus

correlates with decreasing naphthalene or aromatic contents and can be seen as a unified measure attributed to the sooting propensity of the fuels. Therefore, we show AEI versus the fuel's hydrogen content, which is also used in models to calculate the engine's nvPM particle emissions for specific thrust settings (Teoh et al., 2022b). Of all fuels, the 100% HEFA-SPK has the lowest naphthalene, aromatics and sulfur content and at the same time the highest hydrogen content. Therefore, apparent ice particle emissions from this fuel/engine combination are the lowest in the set of compared fuels. For the Jet A-1 from ECLIF3

on the other hand, it is no surprise that its AEI values are lower than those of the Ref2 fuel as the Jet A-1 has a much lower naphthalene, aromatics and sulfur content. However, it becomes more interesting how the ECLIF3 Jet A-1 performs compared to the SAF blends SSF1, SAF1, and SAF2. Focusing on the fuel constituents mainly responsible for soot and volatile particle formation, naphthalene, aromatics and sulfur, ECLIF3 Jet A-1 had a lower naphthalene content but higher aromatics content compared to SSF1 and SAF1 and lies between those two fuels regarding sulfur content. However, ECLIF3 Jet A-1 has lower

AEI compared to both fuels, indicating that the Rolls-Royce engine leads to reduced nvPM particle emission indices compared to the older IAE V2527 engine. Finally, ECLIF3 Jet A-1 has a higher naphthalene content, more aromatics and more sulfur than SAF2 with at the same time lower AEI. The same relation holds true for the two SAF blends SSF1 and SAF1 where SSF1 has higher naphthalene, aromatics and sulfur content but also lower AEI values. This shows that considered fuel constituents alone are not the only parameters that influence soot and ice crystal formation. Soot formation is strongly dependent on engine

cycle, combustion parameters and combustor design. The Trent XWB-84 engine installed on the Airbus A350 during ECLIF3 is a latest generation engine on a latest generation aircraft compared to the IAE V2527-A5 engine installed on the Airbus A320 ATRA. This is also seen in the ICAO engine emissions data base, which delivers engines emission indices from different en-



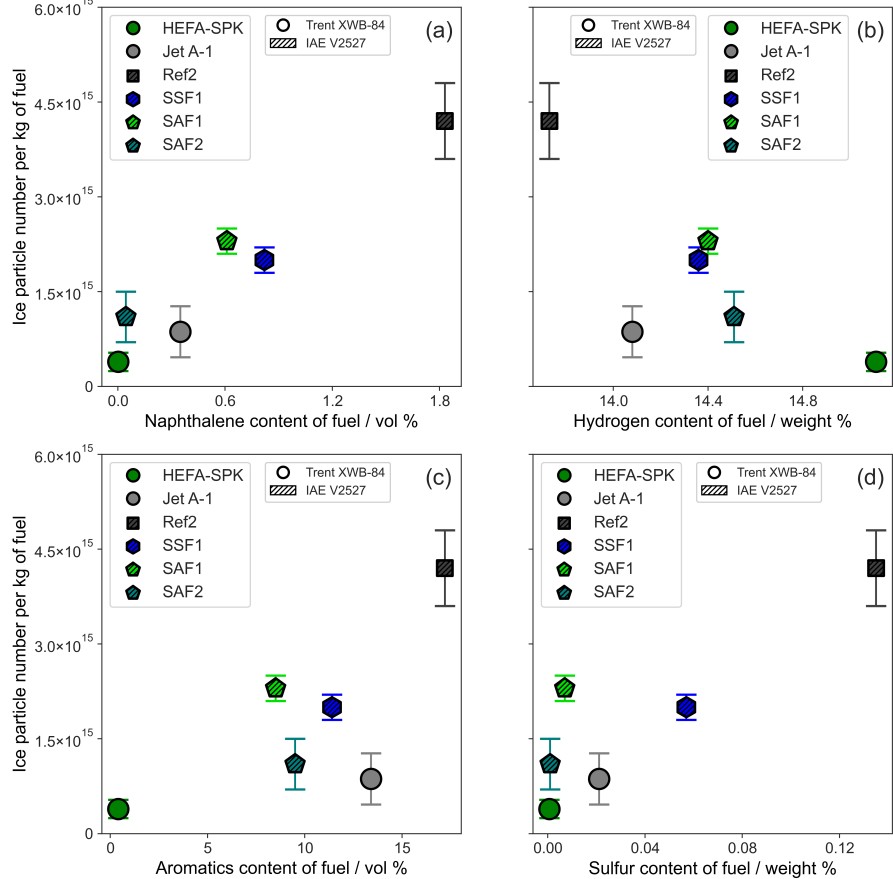

**Figure 4.** AEI of ECLIF1, ECLIF2/NDMAX, and ECLIF3 campaigns versus fuel parameters (a) naphthalene content, (b) hydrogen content, (c) aromatics content, and (d) sulfur content. ECLIF3 fuels (HEFA-SPK, Jet A-1) were burned in a Rolls-Royce Trent XWB-84 engine (circles) while ECLIF1 and ECLIF2/NDMAX fuels (Ref2, SSF1, SAF1, SAF2) were burned in an IAE V2527 engine (hatched symbols). Note: aromatics (naphthalene) content of HEFA-SPK is based on GCxGC measurements (%w/w) and not ASTM D6379 (D1840) due to being below the detection limit of these methods. y-axis error bars are standard deviations of means of the measurements shown as symbols.

gines probed at four thrust settings for the landing-take off cycle. While to some extent, correlations between fuel constituents and apparent contrail ice particle emissions are suggested, other parameters such as engine cycle, combustion parameters, com-

bustor design and atmospheric conditions may influence soot emissions as well. A direct cross-campaign comparison therefore can give hints on trends but differences in ambient conditions and measurement setups impede a direct comparison based solely on the fuel effect as is done for HEFA-SPK and Jet A-1 in ECLIF3. Here, the reduction of 56% from the rigidly reduced data set within one flight can be seen as the reduction potential when a flight is conducted with 100% HEFA-SPK instead of Jet A-1. However, an important fact to consider when interpreting reductions is the reference fuel to which a sustainable aviation fuel or

blend is compared. The Jet A-1 used as a reference fuel in ECLIF3 was relatively clean by measures of naphthalene, hydrogen,



aromatic, and sulfur content compared to the Ref2 fuel and even compared to the blends by some of the fuel properties. A comparison of ECLIF3 HEFA-SPK to Ref2 from ECLIF1 would lead to higher reduction of soot and ice particles simply due to the higher emissions from Ref2 fuel.

## 3.4 Variability in ice particle size distribution

Besides the fuel/engine dependent reduction of overall contrail ice particle numbers, we investigate the variability of the ice particle size distributions in contrails and relate this to the vertical distance to the contrail producing aircraft to account for vortex descent. This study extends beyond the fuel effects on apparent ice emission indices and aims to provide a deeper insight into ice particle microphysics encountered during ECLIF3 contrail measurements. During the contrail vortex regime, 495 exhaust is entrained in the two counterrotating vortices, which propagate downward below the flight level. These vortices produce a wake into which some of the exhaust is detrained at altitudes above the primary wake (Gerz et al., 1998). To avoid measuring particles that sublimate in the descending primary wake, we focused our measurements to flight altitudes in the secondary wake at +96/–48 m vertical distance to the engine of the A350 aircraft. For this purpose, particle size distributions (PSD) of single encounters of contrails formed on Jet A-1 emissions are viewed depending on the difference of detection 500 altitude to emission altitude ($\Delta z$) and shown in Figure 5. For the analysis of ice particle size distributions, we concentrate on contrail encounters from Jet A-1 emissions, as a larger $\Delta z$ is covered and no significant differences in size distribution could be attributed to the different fuels. Due to a better size resolution, we focus on CAS PSDs in ice supersaturated conditions.

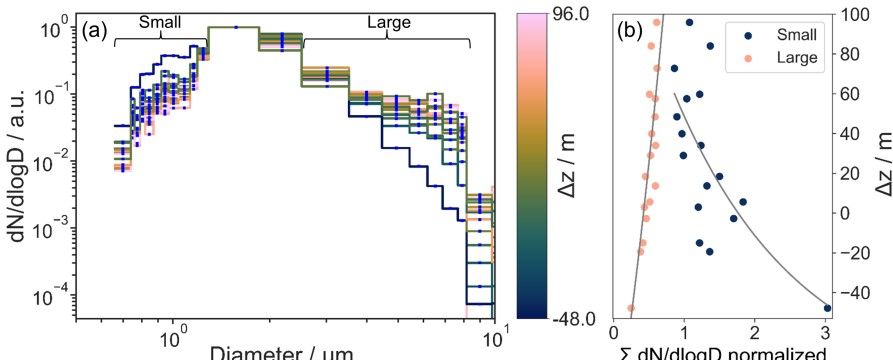

**Figure 5.** Ice particle size distributions (PSD) of contrail crossings from Jet A-1 emissions measured by the CAS-DPOL instrument. The color bar shows the difference of detection altitude to the emission altitude ($\Delta z$) in meters with positive values (negative values) defined as detected above (below) the emission altitude. Panel (a) shows the PSDs in number concentrations per logarithmic bin width normalized to the bin with highest concentration. Bins are classified as "small" sizes and "large" sizes and are marked as such. Panel (b) shows the summed bins of "small" and "large" particle sizes of the normalized PSDs versus $\Delta z$. Gray lines are fits of the exponential function $f(x) = a \cdot b^x + c$ with $a$, $b$, and $c$ being the variable parameters in order to illustrate the trends.



For this study, a mean PSD is calculated for every contrail encounter and the corresponding mean $\Delta z$ is indicated as a color in Figure 5 (a). The PSDs are normalized to the respective size bin with the highest number concentration in order to be able

to compare size ratios independent of absolute number concentrations. As ice particle sizes in a few-minute aged contrails are typically below 10 µm and the contribution of larger ice particles was negligible, the PSDs are shown for sizes up to 10 µm. On the y-axis is the number concentration normalized to the logarithmic width of the respective size bin, which allows comparison of concentrations over various bin widths. While the majority of ice particle sizes lies in the range of 1.3 µm to 2.5 µm, ice particles with sizes below 1.3 µm down to 0.66 µm are henceforth classified as "small" and ice particles with sizes above 2.5

µm up to 8.2 µm are classified as "large". The highest concentration of small ice particles is measured in contrails encountered at large negative distances, hence below the emission altitude and this concentration gradually decreases with increasing $\Delta z$. To illustrate this more clearly, the bins of the respective "small" and "large" ice particle size areas are summed up for every contrail encounter. These normalized distributions are plotted against the respective $\Delta z$ in Figure 5 (b) and facilitate in relating small ice particle size concentrations to large ice particle size concentrations. Relative to the bin with highest concentration,

there are many small and few large particles for low altitudes below emission altitude. The share of small ice particles decreases with growing $\Delta z$ while the share of large ice particles grows.

From this analysis it becomes clear that in this case, more small ice particles are found below the emission altitude, while the occurrence of large ice particles grows with increasing $\Delta z$ within the secondary wake. Ice particles detected at different $\Delta z$ have experienced different temperatures and humidity values along their trajectories from formation of ice in the jet phase and

subsequent vortex processing, leading to the variations in ice particle size distributions. While there is no discernible trend of AEI depending on $\Delta z$ for RHi > 100%, there appears to be a linear reduction of total ice particle concentration with growing $\Delta z$ when the summed dN/dlogD of all bins is regarded as shown in Figure A1. The measurements thereby confirm simulations by Paugam et al. (2010) and discussed by Paoli and Shariff (2016), which predict the location of largest ice crystals at the top of the secondary wake. There, fewer particles compete for the available water vapor thereby allowing the growth of larger

particles while water vapor is distributed among more particles at the higher ice crystal number concentrations found at lower $\Delta z$ resulting in smaller ice crystals. Unterstrasser (2014) finds from large eddy simulations (LES) that number concentration distributions in vertical contrail profiles of five-minute old contrails are non-symmetrical with concentrations decreasing more rapidly for altitudes above emission level compared to altitudes below emission level. Similar results were found in observations (Jeßberger et al., 2013; Schumann et al., 2013). Consistent with these findings, we observe that total number concentrations

are systematically lower for $\Delta z \geq 40$ m. We find contrail ice crystals up to 96 m above the flight altitude of the source aircraft, which can be explained by uplift of ice crystals in the secondary wake and the primary vortex caused by vertical oscillation of the plume interacting with the stratified ambient atmosphere (Brunt-Väisälä dynamics). In addition, the adiabatic increase in temperature during the vortex related descent contributes to the decrease in ice particle size with increasing distance below the emission altitude.

The ice particle size distributions also show for most encounters in the secondary wake that very small ice particles with sizes below 1 µm contribute little to the total size distribution. Similar results are found by Voigt et al. (2021) for the semisynthetic jet fuel with lower AEI values, while higher AEI values for the Jet A-1 lead to smaller ice particle sizes due to more initial ice





particles competing for the same amount of water (ambient and from the engine) and thus stay smaller. As AEI values for both fuels in ECLIF3 are even lower than AEI values of the semisynthetic jet fuel investigated in Voigt et al. (2021), the similar

PSDs for Jet A-1, HEFA-SPK and the semisynthetic jet fuel are consistent with the conclusion of larger particles correlating with lower AEI values. Schumann et al. (2013) investigate the emission index (EI) profiles of trace gases, aerosol and ice crystals in the normalized wake vortex coordinates behind small medium and large aircraft. They found far higher ice particle concentrations in the upper contrail parts than in the descending primary vortex, while passive tracers showed opposite trends. Their ice particle contrail profiles are similar to the present results for large ice particles shown in Figure 5 (b).

**3.5 Comparison of observed and CoCiP modeled apparent ice number emissions indices**

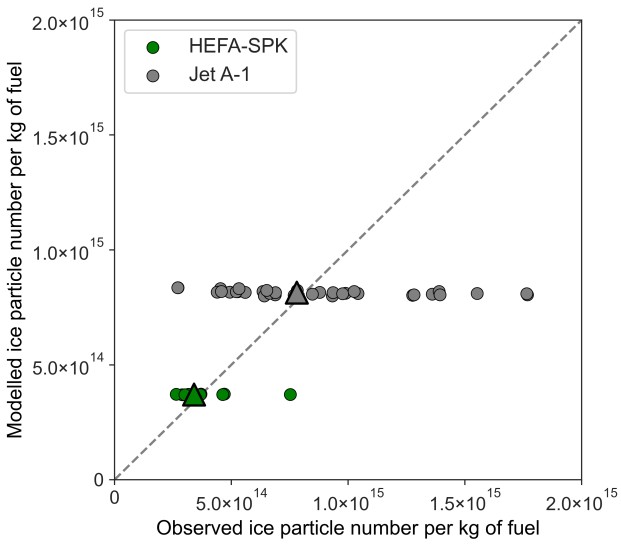

**Figure 6.** Comparison of observed apparent ice emission indices to results from the CoCiP model. Triangles represent median values of the respective fuel and the dashed line shows the ideal 1:1 relationship between observation and model.

The CoCiP model was used to calculate the concentrations of contrail ice particles and emissions trace gases in the contrail plume with vertical and lateral Gaussian concentration profiles expanding with time. The initial dilution in this model can be adjusted to values representative for the primary vortex or the secondary wake based on initial contrail plume depth and width as described in Schumann (2012) and based on the approximate interpolation law described in Schumann et al. (1998). Here we

use an initial dilution value representative for the secondary wake measurements in agreement with the measurements, thereby offering a better representation of the dilution-dependent ice particle loss. The Gaussian profile for both, ice and tracers, implies that the apparent emission indices AEI for ice particles are uniform over the contrail cross-section, while in reality it varies with height and lateral separation of the measurement position relative to the plume center. This limitation could be overcome only by using a more complex plume mixing model. Figure 6 shows AEI values observed during ECLIF3 measurements and

their corresponding values as predicted by the CoCiP model for the two fuel types considered. The plotted observations are



reduced to a data set as described in section 3.1 and therefore only contain contrail encounters in supersaturated ambient conditions at stable engine conditions. As a consequence of the uniform AEI profile in the contrail cross-section, the CoCiP model predicts only a narrow range of AEI values for each fuel while a far wider range of AEI is measured. The reduction in ice particle number concentrations with increasing H-content of fuels is well represented in the CoCiP model, which starts from the model-computed soot emission index ($8.2 - 8.7$ ($\times 10^{14}$ kg$^{-1}$) for Jet A-1 and $3.8 - 3.9$ ($\times 10^{14}$ kg$^{-1}$) for HEFA-SPK), independent of the measured emissions, and in this respect the model is fully consistent with the observations. Ice particle numbers are only slightly reduced compared to the initialized soot emission indices, indicating very little ice crystal loss in the secondary wake contrail measurements. This is reasonable for the data set in supersaturated ambient conditions, which has been chosen to increase comparability of ice particle numbers with as little ice particle loss as possible. The assumption that ice particles and passive tracers (NO$_y$, NO$_x$ or CO$_2$) mix with ambient air homogeneously is not perfectly satisfied in reality. Additionally, the passive tracers are inert on the time scales considered here, while ice particle concentration reductions due to ice particle loss are influenced by the level of dilution. Different measurement positions in the contrail coincide with different states of entrainment of ambient air and resulting states of ice crystal processing, thereby resulting in a wider distribution of AEI values. The modelled and observed median AEIs for Jet A-1 and HEFA-SPK fuels match almost perfectly, see the 1:1 relationship in Figure 6. This illustrates that the CoCiP model is able to accurately predict average apparent ice emission indices of contrails for the investigated fuels and given conditions while small-scale fluctuations are not resolved. The trend of reduced apparent ice emission indices for ultra-low aromatic HEFA-SPK fuel compared to conventional Jet A-1 is captured by the CoCiP model, which reflects the experimental findings well. The results are also in line with simulations by Teoh et al. (2022b), who calculate the effect of enhanced hydrogen content fuels on soot particle emissions, ice crystals in contrails and the related radiative forcing.

### 3.6 Impact of reduced soot particle numbers on the radiative forcing of contrails

Changes in initial ice crystal numbers can have a large impact on the microphysical processes within the contrails, on the contrail life cycle and optical depth (Unterstrasser and Gierens, 2010; Jeßberger et al., 2013; Schumann et al., 2013; Bier et al., 2017) and consequently on contrail cirrus radiative forcing (Bier et al., 2017; Burkhardt et al., 2018; Bock and Burkhardt, 2019; Teoh et al., 2022b). Here we investigate the effect of lower soot particle number emissions due to the use of 100% HEFA-SPK combined with a lower fuel sulfur content on the radiative forcing of contrail cirrus. As explained in section 2.2.6, we simulate contrail cirrus using an estimate for the global average Jet A-1 soot number emission index and in a second experiment we prescribe a lower soot number emission index in line with the measured reduction in AEI of around 60%. This assumed soot reduction is higher than the measured soot reduction of 35%. However as discussed in 3.3, during the ECLIF3 campaign the reference Jet A-1 fuel was a relatively clean fuel regarding fuel constituents that increase sooting propensity such as aromatics and naphthalene. In order to gain insight into the soot reduction potential for a realistic global mean as opposed to an extreme case, current fleet average soot emissions based on Teoh et al. (2023b) were chosen as reference. As the reduced soot particle activation by the low sulfur HEFA-SPK (Wong and Miake-Lye, 2010) as well as volatile particle activation by e.g. sulfur



as described in Section 3.2 is not covered in ECHAM5-CCmod, we translated the measured AEI reduction directly into a
reduction of emitted soot particle numbers. The simulation parameters are summarized in Table 2.

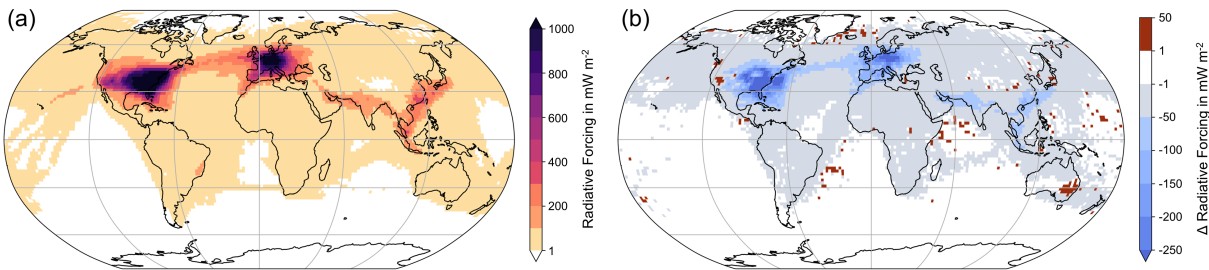

**Figure 7.** Global ECHAM5-CCmod model results of (a) radiative forcing for global mean reference soot emissions for conventional fuel use and (b) changes to radiative forcing resulting from reduced particle emissions for 100% HEFA-SPK.

The estimate for contrail cirrus radiative forcing when using Jet A-1 fuel is $72\,\mathrm{mW/m^2}$ for the year 2018. This estimate lies in the lower part well within the range of the contrail cirrus radiative forcings (33 to $189\,\mathrm{mW/m^2}$) given in Lee et al. (2021). Figure 7(a) depicts that the global estimate is dominated by the main air traffic areas, namely the United States and Europe. As a consequence, these regions also show the largest absolute reduction in radiative forcing when reducing the number of emitted

soot particles as shown in Figure 7(b). Nevertheless, they still remain by far the largest contributors in the 100% SAF case. The reductions in contrail cirrus radiative forcing are larger in the northern parts of the main air traffic areas of Europe and Northern America, which is in line with Bier and Burkhardt (2022) who find a smaller impact of soot number reductions on ice crystal number concentrations over Florida than over the areas slightly further to the north and a similar but weaker gradient over Europe. This is also in line with the dependency of contrail ice nucleation on ambient temperatures that lead to lower

ice nucleation at warmer temperatures and therefore a weaker dependency on soot number emissions. Significant reductions in contrail cirrus radiative forcing can be also found over Southeast Asia. The global mean radiative forcing estimate for the 100% SAF run is $53\,\mathrm{mW/m^2}$ for the year 2018. Hence, following the ECLIF3 measurements and reducing soot number emissions in line with measurements for a pure SAF fuel on the global scale, a decrease in contrail cirrus radiative forcing of 26% may be achieved. Slight fluctuations in the change in radiative forcing shown for a few areas in Figure 7(b) are due to differences in

the natural variability in the atmospheric fields between the two simulations.

The decrease of contrail cirrus radiative forcing of 26% for a decrease in soot number emissions of 60% is in line with the estimates of Burkhardt et al. (2018) and Bier and Burkhardt (2022) who find a non-linear dependency of contrail cirrus radiative forcing when reducing soot number emissions. This dependency originates on the one hand from the non-linearity in cloud

microphysical processes and on the other hand from saturation processes. Besides the number of ice crystals in contrail cirrus clouds, their ice water content and their temporal evolution controlling life time and coverageare important parameters when estimating contrail cirrus radiative forcing. In the main air traffic areas, despite significant reductions in contrail ice crystal numbers the ice water content of contrail cirrus clouds is nearly unchanged when using 100% SAF relative to using Jet A-1





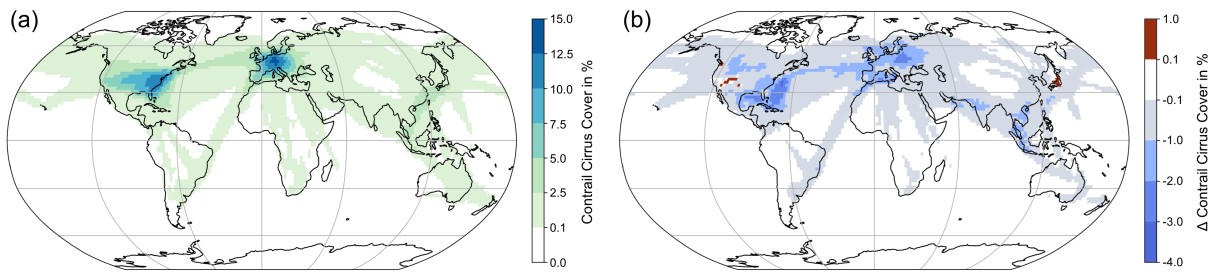

**Figure 8.** Global ECHAM5-CCmod model results of (a) contrail cirrus cover for global mean reference soot emissions for conventional fuel use and (b) changes to contrail cirrus cover resulting from reduced particle emissions with the assumed use of 100% SAF.

(not shown). In order to reduce the ice water content of contrail cirrus earlier in the contrail cirrus life cycle, significantly lower

ice nucleation would be needed such that ice crystals on average grow faster, resulting in earlier sedimentation. In contrast, the contrail cirrus coverage of clouds with an optical depth of >0.05 decreases in particular downwind of the main air traffic areas when reducing contrail ice nucleation (Figure 8(b); cf. Figure 8(a)). In those areas, in which many aged contrails can be found, contrail optical depth is lower and lifetimes are shorter so that contrail cirrus coverage is reduced. Hence, the clouds are not advected as far downwind in the 100% SAF run as in the reference simulation.

We have simulated RF due to a reduction of the soot number emission index by 60%, consistent with the measured reduction in the apparent ice number emission index. As explained above, the reduction in ice crystal numbers is not only dependent on the SAF fuel but also on the reference Jet A-1, which was a relatively clean fuel within the ECLIF3 campaign. Assuming the ECLIF3 HEFA fuel and a slightly "dirtier" Jet A-1 would have led to a slightly larger decrease in contrail cirrus RF. Assuming a reduction in soot number emissions of 66% would have led to a reduction of contrail cirrus RF by about 30% (Bier

and Burkhardt, 2022). León and Lee (2023) calculate a similar reduction in contrail radiative forcing using radiative transfer simulations for two particle size distribution schemes. For the northern Atlantic flight corridor and air traffic in 2019, (Teoh et al., 2022b) find a 43% reduction in contrail radiative forcing and a 51% soot particle reduction, calculated based on the fuel hydrogen content of 100% SAF. This lies within the uncertainties in contrail radiative forcing of current model estimates (Lee et al., 2021).

**4 Conclusions and outlook**

In the course of the ECLIF3 campaign, a measurement flight conducted on April 2021 was identified as having suitable conditions for a side-by-side comparison of contrail properties from $100\%$ HEFA-SPK sustainable aviation fuel and a conventional reference Jet A-1 fuel. During this flight, ice crystals, together with the trace gases $CO_2$ and $NO_y$, as well as aerosols and water vapor were measured in situ behind a long-range Airbus A350-941 equipped with latest-generation Rolls-Royce Trent

XWB-84 engines. Using data from the two forward-scattering laser spectrometers, CAS and CAPS, onboard the DLR Falcon research aircraft, apparent ice emission indices for the two probed fuels were derived. Thereby for similar atmospheric and



engine operation conditions of the preceding aircraft within a single flight, a reduction of ice number concentrations of 56%
for near zero aromatic and near zero sulfur HEFA-SPK compared to Jet A-1 were observed while nonvolatile particle emissions
were reduced by 35%. These reductions were found to depend on the fuel composition. In particular, the lower sulfur content of
the HEFA-SPK might explain the stronger reduction in ice crystals compared to the soot reduction. Also, an influence of engine
cycle, combustion parameters, combustor design and atmospheric conditions becomes apparent when comparing ECLIF3 AEI
to results from previous campaigns where an older IAE V2527 engine with higher soot particle emissions had been probed
(Voigt et al., 2021). Fuel composition of Jet A-1 and SAF are variable in terms of their hydrogen, aromatic and sulfur contents,
which impact particle emissions and should be taken into account for the decision on strategies to reduce the climate impact
from aviation by the use of sustainable aviation fuels. Also, cleaner jet fuel with a naturally (or artificially achieved) low aro-
matic and naphthalene content as well as a low sulfur content could reduce the contrail impact on climate.

Ice crystal particle size distributions were investigated for contrail encounters and a clear dependence of particle size distribu-
tion on the difference of the detection altitude to emission altitude $\Delta z$ was found. On average, larger particles were found up
to 96 m above the emission altitude and smaller particles below. In all cases, the contrail ice crystals had equivalent spherical
diameters of 1.3 µm to 2.5 µm. Here, the experimental data highlight the sensitivity of the ice crystal size distribution to the lo-
cation within the contrail. The contrail cirrus model CoCiP was applied to compute apparent ice particle emission index values
for fuels with a higher hydrogen content and modelled soot emission indices with a one-dimensional Gaussian plume mixing
model. The computed AEI values show less variability compared to the measurements but very similar median values. Hence,
the model is able to capture the measured soot and ice particle reductions based on the fuel hydrogen content parameterization.
In order to assess the mitigation potential of the climate impact from contrails by the use of 100% SAF, we performed global
model simulations applying experimentally derived ice particle reductions. The results suggest a reduction of approximately
26% in contrail radiative forcing for a 60% reduction in soot number concentrations by the use of 100% SAF, applied to the
global fleet average for the year 2018. These reductions are slightly lower than previous model predictions (Burkhardt et al.,
2018; Teoh et al., 2022b) but well within model uncertainties for contrail radiative forcing. Absolute reductions were largest
over the main air traffic areas of Europe and the USA with slightly lower reductions over the southern parts of the main air
traffic areas. Contrail cirrus coverage was predominantly reduced downwind of the main air traffic areas, which primarily con-
tain aged contrail cirrus, in line with a reduction in the contrail cirrus lifetimes (Burkhardt et al., 2018).

The in situ measurements of contrails at cruise altitudes provide insight into the potential benefits of the use of 100% SAF
compared to fossil Jet A-1 fuels at cruise conditions for the current fleet. While measured HEFA-SPK provides a benefit re-
garding the climate impact from contrails, the total climate benefit of different type of SAFs depends on the method of fuel
production and the type of SAF used. In addition, also the aromatic composition of the kerosene plays a role. With currently
limited quantities and higher monetary costs of SAF compared to fossil fuels, one approach could be to preferentially replace
the "dirtier" Jet A-1 fuels containing high naphthalene, aromatics, and sulfur content with SAF. Another approach could be
to try to achieve "cleaner" Jet A-1, pending increased availability of SAF. Further, approaches such as intelligent rerouting
of flights together with a targeted use of SAFs on routes with a high probability of persistent contrails could be pathways to
maximize effectiveness as long as SAF is a limited resource (Burkhardt et al., 2018; Teoh et al., 2022b). Finally, a complete




life-cycle analysis is necessary for every individual fuel in order to evaluate its $CO_2$ footprint and its non-$CO_2$ effects in order to assess a flight's total climate impact.

*Data availability.* The data are collected at the HALO database at https://halo-db.pa.op.dlr.de/mission/124 (doi: 10.17616/R39Q0T)

## Appendix A:  Details of ice particle number concentration determination

Based on wind-tunnel measurements and computational fluid dynamics (CFD) simulations, it has been shown that the Falcon's true air speed (TAS) is a more accurate measure of the sample air speed (SAS) inside the sampling tube of the CAS-DPOL and CAPS-DPOL instruments than the particle air speed (PAS) measured by the respective instruments. The Falcon's TAS is therefore used as the sample air speed leading to a calculation of ice particle number concentrations based on:

$$N = \frac{n}{\text{TAS} \cdot \text{SA} \cdot \Delta t}, \tag{A1}$$

where $n$ is the sum of particle counts passing through the sample area SA within the respective sampling interval $\Delta t$. The use of TAS has been characterized with an uncertainty of 7%. Coincidence effects, which can lead to undercounting and simultaneous oversizing for higher particle concentrations, have been corrected according to a coincidence correction function, leading to a coincidence corrected number concentration:

$$N_{\text{coinc}} = \frac{N \cdot 6323}{6527 - N}, \tag{A2}$$

where $N$ and $N_{\text{coinc}}$ are in $\text{cm}^{-3}$. Although this correction was established for the CAS instrument, it is assumed to be a reasonable approximation for the CAPS as well.

Further, CAS and CAPS counting statistics uncertainty is characterized by Poisson distribution uncertainty for every contrail encounter (Baumgardner et al., 2017). The described uncertainties are propagated using Gaussian error propagation leading to a mean number concentration measurement uncertainty of $19.8\%$ for the CAS and $21.5\%$ for the CAPS.

## Appendix B:  Selection of data sequences for the derivation of apparent ice emission indices

The difficulty of drawing any reliable conclusions from time series illustrates the need for the dilution-corrected apparent contrail ice emission index and not all of the collected data shown in Figure 1 can be used to calculate valid apparent ice emission indices. Sequences where the Falcon was not fully immersed in the contrail lead to a strong mismatch between ice particle concentrations on the one side and trace gas and nvPM concentrations on the other, an example of which is demonstrated from 11:30:00 UTC to 11:33:00 UTC where ice particle number concentrations are very low while trace gas and nvPM concentrations are at similar levels as in the previous parts of the Jet A-1 sequence. Together with strongly differing time series curve shapes, a preferential immersion of the Falcon fuselage top, where the aerosol and trace gas inlets are located, compared to only partial immersion of the wing-mounted ice particle instrumentation is suggested. As a result, the





ice particle measurement in this sequence is classified as invalid. Further, for a valid comparison of AEI it has to be ensured that ice particles are in a quasi-stable state and that ice number concentrations do not change significantly on measurement time scales by sublimation. As ice crystals form in ice supersaturated conditions and are measured about a minute later at ice supersaturated conditions in the secondary wake, we hypothesize that no ice crystal loss has occurred, e.g. due to vortex loss processes (Unterstrasser, 2014). By only further analyzing data in ice-supersaturated conditions with RHi > 100%, this focus

on quasi-stable ice particle numbers is achieved. Finally, only data is chosen where the engine was determined to have operated in a stable condition at time of emission, defined as emission sequences where high pressure compressor outlet temperatures T30 stay within 4 K of stable T30 values. After filtering data according to the above described criteria, a data set remains with quality-controlled valid data points, which can be used to calculate comparable AEI values. This reduces the sampling times from the original 750 s to 183 s for Jet A-1 and from 522 s to 123 s for HEFA-SPK.

**Appendix C: Number concentration values of Jet A-1 size distributions**

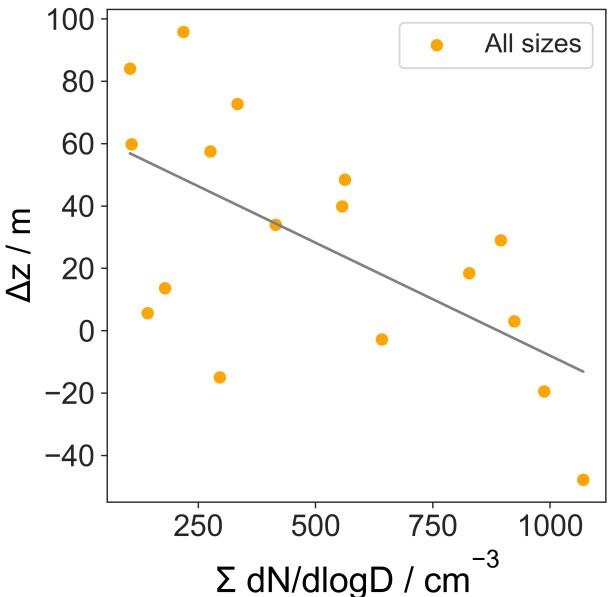

**Figure A1.** Summed bin-normalized number concentrations dN/dlogD of all sizes from contrail encounters shown in Figure 5 together with a linear fit function in gray.

In order to assess the total number concentrations of contrail sequences, the summed bin-resolved dN/dlogD are calculated and plotted against the difference to emission altitude $\Delta z$ in Figure A1. The values are calculated to illustrate trends in the shown particle size distributions also given in dN/dlogD and do not reflect absolute number concentrations N. The trend of growing ice particle number concentrations with smaller $\Delta z$ is indicated by a linear fit function in gray.



*Author contributions.* CV, SK, CR, and AD planned and coordinated the flight experiment. RM, DS, RD, SK, TH, VH, AM, AR and MS performed the in-flight measurements and analyzed the data. RM performed the contrail ice data evaluation and wrote the paper. CV conceptualized the study. CWR and UB performed the climate model calculations and authored the respective sections. US performed a contrail model comparison and authored the respective sections. MG assisted in aircraft data analysis. PS, PM, and DL performed $NO_x$ model analysis and provided $EI_{NO_x}$ data. RS provided the fuels and fuel analysis. TS and PLC performed ground measurements and fuel analysis. All authors contributed to the paper.

*Financial support.* This research has been supported by the Deutsche Forschungsgemeinschaft within SPP 1294 HALO under contract no VO 1504/7-1 and VO 1504/9-1.
The article processing charges for this open-access publication were covered by the German Aerospace Center (DLR) within DEAL.

*Competing interests.* Authors CR and MG are employed by Airbus Operations, authors PS, PM and DL are employed by Rolls-Royce plc. and author RS is employed by Neste Corporation. All other authors declare that they have no conflict of interest.

*Acknowledgements.* The Scientific colour map batlow (Crameri, 2021) is used in this study (Figure 5) to prevent visual distortion of the data and exclusion of readers with color vision deficiencies (Crameri et al., 2020).



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
