# Peer review of "Powering aircraft with 100% sustainable aviation fuel reduces ice crystals in contrails"

_EGUsphere, 2023_

## Referee Comment (RC1)

**General comments:**

In this paper, the authors discuss the impact of using 100% sustainable aviation fuels (SAF) on contrail formation, involving measurements of particle emissions and contrails from 100% SAF combustion, compared to Jet A-1 fuel. The results indicate a significant reduction in ice particle numbers per mass of burned fuel when using 100% SAF, suggesting that this could be an effective way to reduce the climate impact of aviation. The research also explores the effects of different fuel compositions on soot and ice particle emissions, as well as their potential impact on atmospheric conditions and climate forcing. The study is methodologically sound, the data is very valuable and is of significant value in exploring the impact of sustainable aviation fuels on reducing aircraft contrail effects. However, I must express three major concerns as well as a few specific issues related to the content, which I will delve into more deeply below. There is no doubt that with necessary revisions, the work will be worthy of publication. Nonetheless, it is imperative to note that major revisions are required to elevate the study to its full potential.

**Three major issues:**
**1. The method of determination of the apparent ice emission index.**
There are two critical assumptions in your methodology that warrant further elucidation or validation: **a) Uniform NOy Mass Across Different Fuels:** The method you proposed utilizes the ratio of ice crystal to NOy concentration to determine the ice crystal concentration per unit mass of fuel burnt. This method's effectiveness is predicated on the assumption that different masses of fuel (e.g., SAF and Jet A-1) produce the same mass of NOy. However, it appears that the manuscript lacks experimental data or theoretical justification to support this assumption. To bolster the persuasiveness of your research, I recommend providing additional evidence or a detailed analysis to validate this key assumption. **b) Assumption of Similar Atmospheric Influence on Ice Crystals and NOy:** Your approach, based on the relative changes in ice crystal and NOy concentrations to offset atmospheric dilution effects, seems to assume that ice crystals and NOy behave similarly in the atmosphere, unaffected by processes like evaporation, growth, or droplet freezing. Given that the formation and transformation of ice crystals in contrails are dynamic and complex processes, this assumption might require further substantiation. Specifically, the evaporation or growth of ice crystals and the freezing of droplets could significantly influence ice crystal concentrations, potentially leading to divergent behaviors between ice crystals and NOy. Thus, I suggest that you further explore the validity of this assumption and consider the potential impacts of these factors on your study's outcomes.

**2. The reduction of ice crystals due to only BC?**
The observed reduction in ice crystal concentration in aircraft contrails, alongside a decrease in Black Carbon (BC) concentrations, is particularly noteworthy. You suggest that this reduction in BC is a significant contributing factor to the observed decrease in ice crystal formation due to the use of SAF. However, it is well-recognized that a range of particulate matter, not limited to BC, can act as ice nucleating particles, particularly at the cold temperatures typical of contrail formation altitudes. Organic aerosols, both volatile

and non-volatile, can also contribute to ice nucleation (e.g. Tian, P., et al. (2022). The manuscript mentions the presence of non-volatile organic aerosol and VOC emissions that can transform into organic aerosols in aircraft exhaust. Considering this, attributing the reduction in ice crystal formation solely to the reduction in BC might overlook the potential role of these other aerosols.

3. The manuscript is based on the data collected from only one flight experiment. Figure 1 display the total observation time was smaller than one hour. While the findings are intriguing, the variability and complexity of atmospheric conditions raise concerns about the representativeness and generalizability of these results. Atmospheric conditions, including temperature, humidity, and aerosol content, can vary significantly and impact contrail formation. A single flight experiment might not sufficiently capture this variability.

**Specific comments:**

1. The manuscript lacks a clear description of the methodology employed for measuring Black Carbon concentrations. Understanding the measurement technique is crucial as different methods can yield varying results, commonly employing a Single Particle Soot Photometer (SP2). However, this manuscript using a thermo denuder at 250 degrees Celsius to measure refractory BC, this might cause bias in BC measurement, as some non-volatile OA could also survived even after 350 degrees Celsius suggested by Hu, K. et al., (2022) and Tian, P. et al. (2022).

2. The CAS instrument is capable of measuring droplet size distributions in the range of 2-50 micrometers. Importantly, it also provides polarization signals from the backward scattering of individual cloud particles, enabling the differentiation between liquid droplets and ice crystals. This feature is particularly relevant to your study as it can offer a more detailed understanding of the phase composition within the contrails. However, the manuscript does not appear to fully explore or utilize this capability of the CAS.

3. The manuscript indicates that comparisons were made with data above the ice supersaturation threshold to minimize atmospheric interference. However, it lacks a detailed explanation of how this supersaturation state was measured or determined. The specifics of measuring such a critical parameter are vital for understanding and replicating your findings.

4. Some sentences could benefit from better punctuation to enhance readability and clarity.

5. Ensure that all references are formatted consistently and according to the journal's guidelines.

**Reference:**
Tian, P., Liu, D., Bi, K., Huang,M., Wu, Y., Hu, K., et al. (2022). Evidence for anthropogenic organic aerosols contributing to ice nucleation. *Geophysical Research Letters*, *49*, e2022GL099990. https://doi.org/10.1029/2022GL099990.

Hu, K., Liu, D. T., et al. (2022). Identifying the Fraction of Core−Shell Black Carbon Particles in a Complex Mixture to Constrain the Absorption Enhancement by Coatings. Environ. Sci. Technol. Lett. 2022, 9, 272−279. https://doi.org/10.1021/acs.estlett.2c00060

---

## Author Comment (AC1)

**Author's response to referee comments on the manuscript "Powering aircraft with 100% sustainable aviation fuel reduces ice crystals in contrails" (egusphere-2023-2638)**

Referee comments are marked by "Cx", "SCx", or "Comment" and author's answers are marked by "Ax", "SC Ax", or "Answer".

**Referee #1 (comments to author)**

**General comments:**
In this paper, the authors discuss the impact of using 100% sustainable aviation fuels (SAF) on contrail formation, involving measurements of particle emissions and contrails from 100% SAF combustion, compared to Jet A-1 fuel. The results indicate a significant reduction in ice particle numbers per mass of burned fuel when using 100% SAF, suggesting that this could be an effective way to reduce the climate impact of aviation. The research also explores the effects of different fuel compositions on soot and ice particle emissions, as well as their potential impact on atmospheric conditions and climate forcing. The study is methodologically sound, the data is very valuable and is of significant value in exploring the impact of sustainable aviation fuels on reducing aircraft contrail effects. However, I must express three major concerns as well as a few specific issues related to the content, which I will delve into more deeply below. There is no doubt that with necessary revisions, the work will be worthy of publication. Nonetheless, it is imperative to note that major revisions are required to elevate the study to its full potential.

**Answer to referee #1**
We would like to thank the referee for their review and appreciate the critical and balanced assessment of the manuscript. We believe the manuscript has benefited from the comments of Referee #1 which we answer below.

**Three major issues:**
**C1. The method of determination of the apparent ice emission index.**
There are two critical assumptions in your methodology that warrant further elucidation or validation: **a) Uniform NOy Mass Across Different Fuels:** The method you proposed utilizes the ratio of ice crystal to NOy concentration to determine the ice crystal concentration per unit mass of fuel burnt. This method's effectiveness is predicated on the assumption that different masses of fuel (e.g., SAF and Jet A-1) produce the same mass of NOy. However, it appears that the manuscript lacks experimental data or theoretical justification to support this assumption. To bolster the persuasiveness of your research, I recommend providing additional evidence or a detailed analysis to validate this key assumption. **b) Assumption of Similar Atmospheric Influence on Ice Crystals and NOy:** Your approach, based on the relative changes in ice crystal and NOy concentrations to offset atmospheric dilution effects, seems to assume that ice crystals and NOy behave similarly in the atmosphere, unaffected by processes like evaporation, growth, or droplet freezing. Given that the formation and transformation of ice crystals in contrails are dynamic and complex processes, this assumption might require further substantiation. Specifically, the evaporation or growth of ice crystals and the freezing of droplets could significantly influence ice crystal concentrations, potentially leading to divergent behaviors between ice crystals and NOy. Thus, I suggest that you further explore the validity of this assumption and consider the potential impacts of these factors on your study's outcomes.

**A1:**

   **a)** The referee points out that $NO_x$ EIs are very similar for the two investigated fuels. $NO_x$ is produced in the engine combustion chamber at high temperatures and $NO_x$ emission have been shown to correlate strongly with the combustor inlet temperature T30 (Lipfert, 1972). The majority of emitted $NO_x$ is of thermal origin as opposed to fuel $NO_x$ resulting from Nitrogen contained in fuel (Blakey, Rye, & Wilson, 2011). While the fuel H/C content can influence peak flame temperature in the combustor and also the combustor exit temperature, the expected influence on NOx EI is very small and within the model uncertainty. For our experiments, T30 is held constant to ensure comparability and $NO_x$ emission indices are based on predictions from the P3T3 model employed by Rolls-Royce. Therefore, based on results from the P3T3 model rather than just an assumption, we do not expect a significant difference in NOx EI for our investigated fuels.

   **b)** The concept of the emission index has been widely used in previous research and relates number concentrations of a particle quantity to an inert (on the investigated time scales) tracer to provide a measure of number concentration considering dilution (Beyersdorf et al., 2014; Moore et al., 2017; Kleine et al., 2018; Voigt et al., 2021; Bräuer et al., 2021). While this concept is based on the assumption that the mixing behavior in the contrail is the same for the tracer (in this case $NO_y$) and ice particles, it is important to note that the emission index concept does not assume similar processing behavior of ice crystals and the used tracer. Rather, apparent ice particle emission indices (AEI) are highly sensitive to the mentioned processes (evaporation, growth or droplet freezing), which influence ice particle number concentrations, while considering the level of dilution and therefore the stage of ice crystal processing. As an example, contrails affected by sublimation would lead to a gradually decreasing AEI with increasing contrail age compared to steady AEI in contrails unaffected by sublimation. To ensure comparability of AEI between the two investigated fuels, it is important to exclude the influence of sublimation which is why we only consider data in ice supersaturated conditions where we can assume that sublimation processes do not significantly influence ice number concentrations.

**C2. The reduction of ice crystals due to only BC?**

The observed reduction in ice crystal concentration in aircraft contrails, alongside a decrease in Black Carbon (BC) concentrations, is particularly noteworthy. You suggest that this reduction in BC is a significant contributing factor to the observed decrease in ice crystal formation due to the use of SAF. However, it is well-recognized that a range of particulate matter, not limited to BC, can act as ice nucleating particles, particularly at the cold temperatures typical of contrail formation altitudes. Organic aerosols, both volatile and non-volatile, can also contribute to ice nucleation (e.g. Tian, P., et al. (2022). The manuscript mentions the presence of non-volatile organic aerosol and VOC emissions that can transform into organic aerosols in aircraft exhaust. Considering this, attributing the reduction in ice crystal formation solely to the reduction in BC might overlook the potential role of these other aerosols.

**A2:** The referee raises the valid concern that particulate matter other than soot could serve as ice nucleating particles, especially volatile and non-volatile organic aerosols. A number of studies (Kärcher & Yu, 2009; Kärcher et al., 2015; Kärcher, 2018) predict an "enhanced activation of ultrafine aqueous particle" in the regime below approximately $10^{14}$ soot particle numbers per kg fuel, which additionally increase with larger differences of ambient temperatures to the Schmidt-Appleman threshold. The mentioned volatile and non-volatile organic aerosols are of heightened interest in conditions with reduced soot emissions as might be the case for modern lean-burn engine technologies. In our specific case with a conventional RQL (rich-quench-lean) engine, soot emissions are well in the soot-rich regime where soot is predicted to be the primary ice nucleating particle.

**C3.** The manuscript is based on the data collected from only one flight experiment. Figure 1 display the total observation time was smaller than one hour. While the findings are intriguing, the variability and complexity of atmospheric conditions raise concerns about the representativeness and generalizability of these results. Atmospheric conditions, including temperature, humidity, and aerosol content, can vary significantly and impact contrail formation. A single flight experiment might not sufficiently capture this variability.

**A3:**
The referee raises another valid point which highlights the challenges associated with in situ measurements which we would like to elaborate on. During the ECLIF campaign period a total of five contrail flights were conducted. In order to ensure comparability of data, we performed rigorous quality control resulting in data reduction based on engine conditions and meteorological parameters. The most important engine parameters are similar compressor outlet temperatures (T30) and fuel flow to ensure a steady and consistent engine operation. Data are filtered to only include measurements above 100% RHi to exclude sublimation effects and to avoid possible erroneous comparisons of contrail encounters affected by different levels of sublimation. This reduced data set is then further condensed into data with the highest level of measurement quality by ensuring a strong correlation of ice number concentration and dilution tracer concentrations time series. In this way, a large dataset was condensed to a small but high quality dataset from which we derive conclusions presented in the manuscript. This study is therefore to be viewed as a case study where we compare the effect of the use of SAF on nvPM and ice particle concentrations and resulting implications for climate, based on data collected in situ at cruise altitudes. A statistical study of varying meteorological and engine conditions would be desirable in the future and would likely produce variations in results, which is however not achievable with the ECLIF3 dataset.

**Specific comments:**
**SC1.** The manuscript lacks a clear description of the methodology employed for measuring Black Carbon concentrations. Understanding the measurement technique is crucial as different methods can yield varying results, commonly employing a Single Particle Soot Photometer (SP2). However, this manuscript using a thermo denuder at 250 degrees Celsius to measure refractory BC, this might cause bias in BC measurement, as some non-volatile OA could also survived even after 350 degrees Celsius suggested by Hu, K. et al., (2022) and Tian, P. et al. (2022).

**SC A1:** The description of nvPM measurements is described in section 2.2.1 where we introduce the measurement techniques and methodology for aerosol measurements during the ECLIF3 campaign. A Single Particle Soot Photometer (SP2) is not used as this instrument is not sensitive below particle diameters of 70 nm and would therefore not measure a large part of aerosol particles typical for aircraft emission at cruise conditions (Moore et al., 2017; Schripp et al., 2018). The referee is correct in their assessment that the use of a thermodenuder at 250°C could in principle lead to a bias by measuring (semi-) volatile particles and classifying them as nonvolatile. However, if volatile particles survive the thermodenuder, their sizes would likely be very small and lie below the detection limit of 14 nm. Additionally, this bias in absolute number concentrations would be present for both investigated fuels and would therefore not influence the reductions investigated between the two fuels. We therefore argue that this effect lies within the given measurement uncertainty.

**SC2.** The CAS instrument is capable of measuring droplet size distributions in the range of 2-50 micrometers. Importantly, it also provides polarization signals from the backward scattering of individual cloud particles, enabling the differentiation between liquid droplets and ice crystals. This feature is particularly relevant to your study as it can offer a more detailed understanding of the phase composition within the contrails. However, the manuscript does not appear to fully explore or utilize this capability of the CAS.

**SC A2:**
The referee raises a good question about the backscattering data recorded by the employed ice particle instrumentation of the presented in situ ice particle measurements. The CAS-DPOL and CAPS-DPOL instruments are able to measure size distributions from 0.5 – 50 µm from forward-scattered data. It is correct that both instruments also are equipped with backscattering detectors for S- and P-polarized light from backscattered P-polarized light. This allows the calculation of a depolarization ratio, which indicates the sphericity of the measured particle. For spherical particles, a depolarization ratio near 1 is expected while values closer to zero are expected with increasing asphericity.

The CAS-DPOL is an older instrument version with only one gain stage while the newer CAPS-DPOL has two gain stages. The gain stage amplification from the older CAS-DPOL is not strong enough to resolve signals from small particles (< 7 µm) typical for contrails and is therefore not suitable for a backscattering analysis of contrail ice particles. The newer CAPS-DPOL is in principle able to resolve the depolarization ratio of contrail ice particles above 1 µm diameter. Here we find for contrail ice particles a broad distribution of depolarization ratios which indicates ice crystals with deviation from sphericity and random orientation. In theory, backscattering signals from liquid particles would blend into the smeared out distribution of depolarization ratios, thereby not being able to differentiate between liquid and ice phase. However, the contrail measurements were taken at ambient temperatures of approximately -60°C, well below the homogeneous freezing threshold of approximately -40°C (Koop & Murray, 2016). We can therefore assume that measured contrail ice particles are in fact ice and not liquid. From the backscattering data, we cannot determine the exact shape or aspect ratio of the ice particles and instead assume rotation ellipsoids with an aspect ratio of 0.75 as specified in the manuscript. As our backscattering data does not contribute any new information about contrail ice particle shapes or asphericity, we decided to instead direct our focus to the forward scattering data based apparent ice emission indices and ice particle size distributions from which we can derive the novel findings presented in the manuscript.

**SC3.** The manuscript indicates that comparisons were made with data above the ice supersaturation threshold to minimize atmospheric interference. However, it lacks a detailed explanation of how this supersaturation state was measured or determined. The specifics of measuring such a critical parameter are vital for understanding and replicating your findings.

**SC A3:** Ice supersaturation is defined as relative humidity above ice exceeding 100%. We use the onboard water vapor mass spectrometer AIMS described in section 2.2.2. to measure water vapor mixing ratios from which relative humidity over ice can be calculated in using DLR Falcon onboard temperature measurements. In order to determine if a contrail encounter occurred in ice supersaturated conditions, the RHi time average over the contrail encounter time is calculated. If this average RHi exceeds 100%, the contrail encounter is accepted for further analysis as having occurred in non-sublimating ice supersaturated conditions. If the average RHi lies below 100%, the encounter is rejected as sublimation might have occurred, which can impede valid comparisons. Following findings of Kaufmann (2014) we argue that ice crystals in plumes classified as ice supersaturated in this way are definitely exposed to an ice supersaturated environment and we can exclude significant particle losses due to sublimation.

**SC4.** Some sentences could benefit from better punctuation to enhance readability and clarity.

**SC A4:** We appreciate the referee's emphasis on language quality and will focus on improving the manuscript's punctuation and readability during the revision.

**SC5.** Ensure that all references are formatted consistently and according to the journal's guidelines.

**SC A5:** Care has been taken to conform with the journal's guidelines on reference formatting. We will recheck our formatting and make sure to have consistent and compliant reference formatting.

**Reference:**

Tian, P., Liu, D., Bi, K., Huang,M., Wu, Y., Hu, K., et al. (2022). Evidence for anthropogenic organic aerosols contributing to ice nucleation. *Geophysical Research Letters*, *49*, e2022GL099990. https://doi.org/10.1029/2022GL099990.

Hu, K., Liu, D. T., et al. (2022). Identifying the Fraction of Core-Shell Black Carbon Particles in a Complex Mixture to Constrain the Absorption Enhancement by Coatings. Environ. Sci. Technol. Lett. 2022, 9, 272-279. https://doi.org/10.1021/acs.estlett.2c00060

**Referee #2 (comments to author)**

This is a timely and very relevant study of the effects of using 100% sustainable aviation fuels in a current technology airplane. The manuscript is well written and clearly describes the work done, the data obtained, and the analyzed results. The team has done previous, related in-air measurement campaigns, so their experimental methodology, equipment choices, and data analysis approach and techniques are all highly refined and state-of-the-art. The report provides useful data on the potential benefits of using fuels that have characteristics like the SAF fuel tested in this study, so is important for the aviation industry in its planning for minimizing its future environmental impacts.

Especially notable in this study are several points made in addition to the main findings. In section 3.3 (lines 447 – 455 and thereabouts) and then again in several places in the conclusions (640 - 645, 655 - 669), the authors emphasize that both engine technology and fuel composition can affect the emissions changes. So, improvements due to SAF usage cannot be quantified solely based on the SAF fuel usage alone, since engine technologies and individual SAF compositions both influence the benefits obtained. But equally important, the unique aspect of SAFs is their bio-source and improvement (decrease) in $CO_2$ emissions. While the lower S and aromatic/naphthalene content (or higher H-content) of most SAFs allows the exploration of the potential improvements due to reductions of non-$CO_2$ impacts, these non-$CO_2$ reductions could also be replicated in fossil-derived jet fuel if appropriate fuel composition changes could and would be made. This is an important point, since full implementation of SAF replacement of jet fuel could take decades, and some non-$CO_2$ benefits could be had sooner with adjustments to fossil jet fuel composition if warranted.

So, I have no real criticism of the manuscript and think it is eminently worthy of publication. Perhaps to emphasize that I have read it thoroughly, I will point out that there is a space missing between "coverage" and "are" on line 612.

I could leave the review at this point, but this is exciting work and I cannot refrain from thinking further about their work and its implications. So, the following points are offered for the authors to consider.

**Answer to referee #2**

We are very pleased with the review the referee has given this manuscript and want to thank for the assessment and recommendation for publication.

**C1:** In section 2.2.3 (210 – 213), the NOx EI is estimated by using an industry proprietary method. Thus, the calculation of other EIs are done using this estimated NOx EI rather than any experimentally measured quantity. So, this is a limitation and the paper predicts an uncertainty of 10%. However, the engine operating conditions are very similar for all the cruise conditions encountered. If the NOx prediction is off by some amount due to uncertainty in the prediction tool, it is likely off by a similar value for all cruise conditions. This is more likely a bias on the analysis rather than an error. For instance, the engine T30 for JetA-1 differs from the HEFA-SPK by 6.5 K, resulting in a 17.4 - 17.5 g/kg fuel shift in the NOx EI. If the prediction tool is off by 10%, both results are shifted similarly, so any analysis of the difference between these two fuels is more dependent on the difference between these two EIs rather than the values themselves. Agreed that the absolute EIs will have this error, but the authors may wish to comment that the differences are less error prone than the error in the absolute magnitude. And differences between the two fuels are the focus of this paper.

**Answer to NOx EI remark (C1)**

The referee remarks that the uncertainty given for NOx EI predictions likely would be systematic, i.e. the predicted NOx EI for each fuel would be shifted in the same direction. This seems to be a reasonable assumption which we have now accounted for and discuss in section 2.2.4. Besides the absolute uncertainty for apparent ice emission indices, we had also mentioned a non-systematic uncertainty, which regards the difference in AEI between both fuels rather than their absolute values. The referee mentions that this difference is the focus of the paper, which we fully agree with.

**C2:** The analysis in section 3.4 is quite interesting. It highlights the level of detail of what is known about the flow-field/microphysics interaction, especially regarding the secondary wake and the contrail particle properties there. Figure 5 is quite illuminating, with 5(b) a very nice way to show a trend that is hard to discern as clearly in the main figure (a). In this regard, the authors may wish to note in more detail the changing shape of the size distribution. While many naturally occurring size distributions tend to have log-normal shapes (perhaps due to a central-limit sort of behavior of many competing random processes), a log-normal shape is not physically imposed. And, in this case of the secondary wake, this is a very clear example of the distribution being NOT log-normal and, in fact, the shape is changing across the range of Δz in the secondary wake (otherwise, if the distribution was always log-normal, with the normalization in (a), the curves would all lie on top of one another). The authors may wish to emphasize that the resulting contrail particle size distributions are not log-normal, which has also been observed in some μphysical modeling as well.

**Answer to lognormal size distribution remark (C2)**

The referee highlights that section 3.4 and figure 5 provide insights into the changing shape of ice particle size distributions with the range of Δz in the secondary wake. It is pointed out that the observed size distributions deviate from ideal log-normal shapes. We agree with the referee's assessment that the shape of the presented ice particle size distributions changes with Δz. While lognormal distributions are found in aerosol size distributions distributions (Moore et al., 2017), this is not necessarily the case for ice particle size distributions (Kleine et al., 2018; Voigt et al., 2021).

However, we would like to point out that the size distributions in the manuscript are shown with double-logarithmic axes and normalized to the bin of highest number concentration in

order to highlight the changes in particle size distribution shape. While this type of representation has its benefits in presenting details at lower number concentrations, it might not be the easiest representation to compare to lognormal distributions. Below in Figure 1 (a) we show the same data but in absolute number concentration values and with a linear y-axis. In this axis representation, a lognormal distribution would appear as a classical Gaussian curve shape and it becomes clear that the size distributions in fact do not deviate too much from the shape of a lognormal distribution. To verify this, we conducted lognormal fits to these size distributions and the $R^2$ coefficients of determination of these fits can be seen in Figure 1 (b). For lognormal fits of all shown size distributions, $R^2$ values of over 0.96 are achieved for all distributions except for the one at lowest $\Delta z$, which also visually deviates the strongest from the other distributions. This size distribution is fitted lognormally with a $R^2$ of 0.93. We can therefore conclude that the shown size distributions have varying deviations from perfect lognormal distributions with $\Delta z$ while they are generally approximated well by lognormal function fits.

[Figure]

Figure 1: (a) Ice particle size distributions (PSD) as shown in the manuscript in absolute values and with linear y-axis. (b) Histogram of $R^2$ coefficient of determination for lognormal fits to the shown PSDs.

**C3:** Still in reference to section 3.4, while this level of detail is very interesting to this reader, it is not clear to me how these details relate to the radiative forcing analyzed and discussed in section 3.6. Are these details of the secondary wake fully considered in the RF analysis? If that was discussed, I missed it. If these details have yet to be fully included, perhaps a few sentences or a paragraph about why that is hard and/or what might be needed to do so in the future would be valuable to understand how 3.4 relates to 3.6.

**Answer to global climate model remark (C3)**

The referee points out that the manuscript does not sufficiently convey if and how the in situ findings on ice particle size distributions are used in the global climate model calculations described in section 2.2.6 and presented in sections 3.6. The measured size distributions in section 3.4 represent a snapshot at a specific young contrail age and intend to point out microphysical variations in the vertical extent of the measured contrails. However, the measurements do not provide a study at a wide range of contrail ages and especially not in aged contrail cirrus, which are especially relevant for contrail radiative forcing. Therefore, the in situ contrail ice size distribution findings were not used in this case for the global climate modelling. Instead, the parameterization estimating the ice crystal loss in the vortex phase is based on LES assuming a typical initial ice crystal size distribution and simulating the

subsequent partial loss of contrail ice crystals due to detailed wake vortex dynamics. This means that the impact of the wake vortex dynamics on the contrail ice crystal number is taking into account the processes described in section 3.4. so that the impact of the wake vortex on ice crystal numbers is captured within ECHAM5-CCMod. We agree that this procedure was previously not mentioned in detail in the manuscript and we have added to sections 2.2.6 and 3.6 to make this more clear to the reader.

**Author's changes to the manuscript**

- Moved the position of Table 2 from section 3.1 to 2.2.6
- Typographical errors corrected
- References updated for articles from preprints to accepted articles
- Referee #1 C1 a): Modified sentence and added reference to 2.2.3 to assert thermal NOx as the main influence to NOx EI
- Referee #1 C1 b): Added sentence to section 2.2.3 to clarify representation of ice particle processing in the AEI metric
- Referee #1 C3: Added to section 4 to indicate limitations of measurement and give outlook to necessary future measurements.
- Referee #1 SC1: sentence added to section 2.2.1 to clarify the mentioned sampling bias of the thermodenuder
- Referee #1 SC3: Added to section 2.2.2 to describe the judgement of ice supersaturation in more detail.
- Referee #2 C1: Modified final sentence in section 2.2.4 to illustrate more clearly the difference in systematic and non-systematic uncertainty. Also, $EI_{NOx}$ prediction uncertainty was removed from non-systematic uncertainty, leading to a new uncertainty value.
- Referee #2 C2: Added a sentence to 3.4 to briefly discuss the deviation from lognormal size distributions.
- Referee #2 C3: Added clarifications and a sentence to sections 2.2.6 and 3.6

**References**

Beyersdorf, A. J., Timko, M. T., Ziemba, L. D., Bulzan, D., Corporan, E., Herndon, S. C., Howard, R., Miake-Lye, R., Thornhill, K. L.,Winstead, E., Wey, C., Yu, Z., and Anderson, B. E.: Reductions in aircraft particulate emissions due to the use of Fischer–Tropsch fuels, Atmos. Chem. Phys., 14, 11–23, https://doi.org/10.5194/acp-14-11-2014, 2014.

Blakey, S., Rye, L., & Wilson, C. W.: Aviation gas turbine alternative fuels: A review., Proc. Combust. Inst., 33, 2863–2885, https://doi.org/10.1016/j.proci.2010.09.011, 2011.

Bräuer, T., Voigt, C., Sauer, D., Kaufmann, S., Hahn, V., Scheibe, M., Schlager, H., Huber, F., Clercq, P. L., Moore, R. H., and Anderson, B. E.: Reduced ice number concentrations in contrails from low-aromatic biofuel blends, Atmos. Chem. Phys., 21, 16 817–16 826, https://doi.org/10.5194/acp-21-16817-2021, 2021.

Kärcher, B.: Formation and radiative forcing of contrail cirrus, Nat. Commun., 9, https://doi.org/10.1038/s41467-018-04068-0, 2018.

Kärcher, B. and Yu, F.: Role of aircraft soot emissions in contrail formation, Geophys. Res. Lett., 36, https://doi.org/10.1029/2008GL036649, 2009.

Kärcher, B., Burkhardt, U., Bier, A., Bock, L., and Ford, I. J.: The microphysical pathway to contrail formation, J. Geophys. Res. Atmos.,120, 7893–7927, https://doi.org/10.1002/2015JD023491, 2015.

Kaufmann, S., Voigt, C., Jeßberger, P., Jurkat, T., Schlager, H., Schwarzenboeck, A., Klingebiel, M., Thornberry, T.: In situ measurements of ice saturation in young contrails, Geophys. Res. Lett., 41, 702–709. https://doi.org/10.1002/2013GL058276, 2014.

Kleine, J., Voigt, C., Sauer, D., Schlager, H., Scheibe, M., Jurkat-Witschas, T., Kaufmann, S., Kärcher, B., and Anderson, B. E.: In Situ Observations of Ice Particle Losses in a Young Persistent Contrail, Geophys. Res. Lett., 45, 13,553–13,561, https://doi.org/10.1029/2018gl079390, 2018.

Koop, T., & Murray, B. J.: A physically constrained classical description of the homogeneous nucleation of ice in water, J. Chem. Phys., 145. https://doi.org/10.1063/1.4962355, 2016.

Lipfert, F. W.: Correlation of Gas Turbine Emissions Data. https://doi.org/10.1115/72-GT-60, 1972.

Moore, R. H., Thornhill, K. L., Weinzierl, B., Sauer, D., D'Ascoli, E., Kim, J., Lichtenstern, M., Scheibe, M., Beaton, B., Beyersdorf, A. J., Barrick, J., Bulzan, D., Corr, C. A., Crosbie, E., Jurkat, T., Martin, R., Riddick, D., Shook, M., Slover, G., Voigt, C., White, R., Winstead, E., Yasky, R., Ziemba, L. D., Brown, A., Schlager, H., and Anderson, B. E.: Biofuel blending reduces particle emissions from aircraft engines at cruise conditions, Nature, 543, 411–415, https://doi.org/10.1038/nature21420, 2017.

Schripp, T., Anderson, B., Crosbie, E. C., Moore, R. H., Herrmann, F., Oßwald, P., . . . Wisthaler, A.: Impact of Alternative Jet Fuels on Engine Exhaust Composition During the 2015 ECLIF Ground-Based Measurements Campaign, Environ. Sci. Technol., 52, 4969–4978. https://doi.org/10.1021/acs.est.7b06244, 2018.

Voigt, C., Kleine, J., Sauer, D., Moore, R. H., Bräuer, T., Clercq, P. L., Kaufmann, S., Scheibe, M., Jurkat-Witschas, T., Aigner, M., Bauder, U., Boose, Y., Borrmann, S., Crosbie, E., Diskin, G. S., DiGangi, J., Hahn, V., Heckl, C., Huber, F., Nowak, J. B., Rapp, M., Rauch, B., Robinson, C., Schripp, T., Shook, M., Winstead, E., Ziemba, L., Schlager, H., and Anderson, B. E.: Cleaner burning aviation fuels can reduce contrail cloudiness, Commun. Earth Environ., 2, https://doi.org/10.1038/s43247-021-00174-y, 2021.